# Recurrent Convolutional Neural Networks Learn Succinct Learning Algorithms

**Surbhi Goel**
Microsoft Research &
University of Pennsylvania
surbhig@cis.upenn.edu

**Sham Kakade**
Harvard University
sham@seas.harvard.edu

**Adam Tauman Kalai**
Microsoft Research
adam@kal.ai

**Cyril Zhang**
Microsoft Research
cyrilzhang@microsoft.com

## Abstract

Neural networks (NNs) struggle to efficiently solve certain problems, such as learning parities, even when there are simple learning algorithms for those problems. Can NNs discover learning algorithms on their own? We exhibit a NN architecture that, in polynomial time, learns as well as any efficient learning algorithm describable by a constant-sized program. For example, on parity problems, the NN learns as well as Gaussian elimination, an efficient algorithm that can be succinctly described. Our architecture combines both recurrent weight sharing between layers and convolutional weight sharing to reduce the number of *parameters* down to a constant, even though the network itself may have trillions of nodes. While in practice the constants in our analysis are too large to be directly meaningful, our work suggests that the synergy of Recurrent and Convolutional NNs (RCNNs) may be more natural and powerful than either alone, particularly for concisely parameterizing discrete algorithms.

## 1 Introduction

Neural networks (NNs) can seem magical in what they can learn. Yet, humans have designed simple learning algorithms, even for binary classification, which they cannot match. A well-known example is the class of parity functions over the $d$-dimensional hypercube, i.e., $d$-bit strings. In that problem, there is an unknown subset $S$ of the $d$ bits, and the label of each example $x$ is 1 if $x$ has an odd number of 1's in $S$. While gradient-based learning struggles to learn parity functions (Kearns and Valiant, 1993) even over uniformly random $x$, row reduction (i.e. Gaussian elimination) can be used to find $S$ using only $O(d)$ examples and $O(d^2)$ runtime.

A tantalizing question is whether a NN *can discover an efficient learning algorithm itself*, thereby learning classes such as parities. We refer to this as *Turing-optimality*, since algorithms can be described by Turing machines. More specifically, we will give an example of a simple NN architecture that achieves Turing-optimality. In particular, this is the first NN architecture that provably discovers a efficient parity learning algorithm in polynomial time. The parity learning algorithm is efficient, like row reduction, requiring $O(d)$ examples and $O(d^2)$ runtime. Our learning architecture would be quite simple to describe with a modern library such as PyTorch. However, we do not expect our specific architecture to be especially good in practice, as the constants in our analysis are much too large to be practical. Nonetheless, it does suggest that the ingredients used in the architecture, especially the combination of recurrent weight-sharing across layers and convolutional weight-sharing within layers, may be useful in designing practical architectures for NNs to learn algorithms.

36th Conference on Neural Information Processing Systems (NeurIPS 2022).

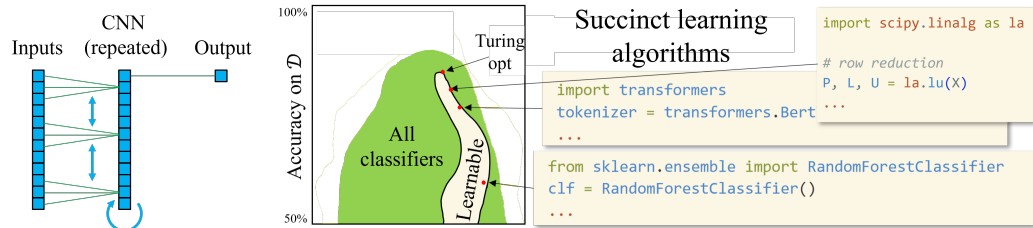

Figure 1: *(Left)* A simple RCNN, in which one small set of parameters is repeated *within each layer* and *across layers*. *(Right)* A Turing-optimal algorithm must output a classifier that is as accurate, on future examples from $\mathcal{D}$, as that which is output by any other succinct efficient learning algorithm. Such classifiers are only a tiny subset of the set of all classifiers, some of which may be more accurate but cannot even be stored in a computer. Moreover, even competing with the best poly-sized NN is intractable, assuming the existence of one-way functions (Kearns and Valiant, 1994); it is not possible for any learning algorithm to efficiently compete with the region depicted in green.

Figure 1 illustrates the difference between *classifiers*, such as NNs, and the learning *algorithms* that learn their parameters, such as Stochastic Gradient Descent (SGD) with a given architecture (we use the term *architecture* broadly to include other algorithmic features including learning random initialization, learning rate schedule, restarts, and hyperparameter search). As *classifiers*, two-layer NNs can compute any Boolean function on $d$ binary inputs, including parity functions. However, it is unclear whether these architectures can learn such functions efficiently using gradient-based approaches without any priors encoded in the architectures.

More formally, a Turing-optimal learning algorithm is one which learns as well as any bounded learning algorithm, specifically a constant-sized Turing machine that outputs a binary classifier in polynomial time, such as row reduction for parity learning. Our key contribution is a simple recurrent convolutional (RCNN) architecture which combines recurrent weight-sharing across layers and convolutional weight-sharing within each layer. The number of weights in the convolutional filter can be very few, even a constant, but these weights can determine the activations of a very wide and deep network. We show that any algorithm $A$ represented by a constant-sized TM has a corresponding constant-sized convolutional filter for which the RCNN computes the same function as $A$. Because the convolutional filter is constant-sized, with constant probability random initialization will find it (or something even better, assuming our reduction is not optimal). Thus, using a validation set and random restarts, the RCNN will find a filter which performs as well as $A$, with high probability.

Unfortunately, the above argument would apply to an RCNN architecture that strangely takes the *entire training set* as input at once, and outputs a classifier. Fortunately, Abbe and Sandon (2020) show how to use a few additional simple NN components and SGD updates to memorize relevant information in the weights of these components. Similarly, we add a few extra non-convolutional layers to our architecture so that it can be learned "normally" with SGD rather than requiring the entire dataset at once. Fortunately, the implementation of this functionality is compatible with the RCNN with only a constant overhead in terms of size.

**Probably Algorithmically Optimal (PAO) learning.**   To define Turing-optimality, it is convenient to formalize a weaker requirement than PAC learning which we call PAO learning, that in some sense turns PAC learning on its head. Rather than requiring optimality among the space of all classifiers $c \in \mathcal{C}$, it requires only optimality compared to classifiers output by learning algorithms $A \in \mathcal{A}$. In many cases $|\mathcal{C}| \gg |\mathcal{A}|$, as $\log |\mathcal{C}|$ and $\log |\mathcal{A}|$ are the number of bits required to encode the parameters (modern models approaching terabytes) and the learning program source code (e.g., kilobytes), respectively (see Arora and Zhang (2021) for a more detailed discussion). In these cases, matching the performance of the classifier output by the best learning algorithm (within some family) may be more reasonable matching the performance of the best overall classifier. Turing-optimality is the special case of $\mathcal{A}$ consisting of the set of succinct programs, specifically constant-sized time-bounded Turing machines. In Section 5, we also discuss how this approach can be used across multiple problems to discover a learning algorithm that can be reused on future problems, so the search need not be repeated for each learning problem.

Like any asymptotic notion, Turing-optimality does not guarantee efficient learning. Just as a polynomial-time algorithm is not guaranteed to be faster than an exponential-time algorithm on inputs of interest, such notions can still provide a useful lens to understand algorithms. If learning algorithm $A_1$ is Turing-optimal and $A_2$ is not, then $A_1$ can nearly match (or exceed) the performance of $A_2$ on any distribution, with polynomial overhead. Data distributions where $A_2$ requires super-polynomially resources to match $A_1$'s performance would need to be examined to see if they are important. We show that a Recurrent Convolutional Neural Network (RCNN) architecture, with random initialization, is Turing-optimal. The contribution of this work is showing that a definition of Turing-optimality is achievable by a simple NN architecture. In future work, it would be interesting to better understand which other combinations of architectures, initializations, and learning rates are Turing-optimal.

## 1.1 Related work

We review some prior lines of work which establish or use other notions of computational universality. We note that most of the notions defined in these works apply to representations rather than algorithms.

**Universal function approximation.** The first line of work relevant to our results is the basic theory of universal function approximation, which quantifies the ability to fit any sufficiently well-behaved function for neural networks (Hornik et al., 1989; Cybenko, 1989; Funahashi, 1989), nearest neighbors (Devroye et al., 1994) and SVMs with RBF kernel (Wang et al., 2004). However, they lack statistical insight, e.g., lookup tables are universal over $\mathcal{X} = \{-1, 1\}^d$ but offer little statistical power. Further refinements (Barron, 1993, 1994; Lee et al., 2017) consider Fourier-analytic criteria for functions to be representable by smaller neural networks. The goal of subsequent lines of work described in this section, as well as the present work, is to investigate the *computationally efficient* approximation of functions—in other words, the ability of neural networks to emulate efficient learning algorithms.

**Turing-completeness of neural architectures.** Siegelmann and Sontag (1995) establish that recurrent neural networks are Turing-complete, using a trick to store the entire TM tape in a single rational number, therefore requiring an extreme amount of bit precision. More recently, Graves et al. (2014) construct a differentiable TM-inspired architecture. A number of recent works establish Turing-completeness (a classical and weaker notion) for variants of the Transformer architecture (Dehghani et al., 2018; Yun et al., 2019; Bhattamishra et al., 2020a,b), motivated by empirical advances in discrete reasoning tasks found in natural language processing, theorem proving, and program synthesis. Recently, Wei et al. (2021) propose a notion of statistically meaningful (SM) approximation which requires the approximation to be statistically learnable as well. They show that Transformer architectures can "SM-approximate" time bounded TMs with sample complexity logarithmic in the time. Unlike our notion, Turing-completeness does not take computational efficiency into account.

**Enumerative program search.** A folklore argument, similar to Levin's classic universal search (Levin, 1973), states that one can achieve Turing-optimality by enumerating all Turing machines of a fixed size, run them all on a training set, and choose the one which performs best on a validation set. The algorithm, however, is also completely infeasible in normal programming languages because the probability of even generating a single program that compiles is minuscule.

**Efficient universality of deep learning.** Most closely related to our work is that of Abbe and Sandon (2020), which shows how, given any circuit $C$, e.g., encoding a learning algorithm for parity, one can initialize the weights of a NN so that it emulates $C$ when the NN is trained by SGD. This emulator requires $C$ to be given as input. Now, row reduction, like any polynomial-time algorithm, can be converted to a circuit $C$. However, the size of this circuit is polynomial in the runtime of the algorithm. This is why $C$ is required as input, e.g., one has no hope of discovering the Gaussian elimination algorithm by random initialization as its probability would be exponentially small in the dataset size. Thus, their algorithm does not "discover" the learning algorithm itself–it is hard-coded into the network. As they discuss, they could encode in the circuit $C$ an enumerative program search, but this is also a parity learning algorithm that needs to be encoded into the network (and is in fact significantly more involved to encode as a circuit). Their work was recently extended to mini-batch SGD by Abbe et al. (2021), and it would be interesting to see if our result could be similarly extended.

## 2 Preliminaries

For simplicity, we focus on binary classification with $\mathcal{Y} := \{-1, 1\}$. For domain $\mathcal{X}$, a (binary) *classifier* is a function $c : \mathcal{X} \to \mathcal{Y}$. For any distribution $\mathcal{D}$ over $\mathcal{X} \times \mathcal{Y}$, the *error* of $c$ is, $\mathrm{err}_{\mathcal{D}}(c) := \mathrm{Pr}_{(x,y)\sim\mathcal{D}}[c(x) \neq y] \in [0, 1]$. A learning algorithm takes as input $m \geq 1$ labeled training examples in $(\mathcal{X} \times \mathcal{Y})^+ = \bigcup_{m\geq 1}(\mathcal{X} \times \mathcal{Y})^m$ and outputs a classifier. For further simplicity, we focus on data on the hypercube $\mathcal{X}_d := \{-1, 1\}^d$. The powers of 2 less than 1 are denoted by $2^{-\mathbb{N}} = \{2^{-i} \mid i \in \mathbb{N}\}$. We say an algorithm is *poly-time* if it runs in time polynomial in its input length, which is $\mathrm{poly}(dm)$ for a learning algorithm when run on $m$ examples in $d$ dimensions.

### 2.1 Turing machines, circuits, and efficient computability

Since our main results require the simulation of an arbitrary efficient learning algorithm, we will need to establish formal notation for relevant concepts from the theory of computation. Various notions of computational efficiency may be used. To be concrete, we may use a 2-tape Turing Machine (TM) where the input is on the first tape and the 2nd tape is used for computation (e.g. see Hopcroft et al. (2001) for a standard reference on Turing machines).

One issue that complicates runtime analysis of learning algorithms is that a classifier may be very slow to evaluate, even if the learning algorithm is fast.[1] There are two solutions to this issue, which are equivalent up to polynomial time. The first is learning algorithms that output classifiers, which we represent as Boolean circuits. Circuits circumvent this technicality because they can be evaluated in time nearly linear in the time it takes to output them. Thus time spent on classification is folded into training time. Moreover, any binary classifier on $\mathcal{X}_d$ can be represented as a circuit, and it is straightforward to convert a NN to a circuit with linear blowup. It is also well-known that other universal representations such as (time-bounded) TMs can be converted to Boolean circuits in polynomial time using unrolling. Other succinct representations could be used, but this choice simplifies runtime analysis.

Formally, we assume that each classifier output by a learning algorithm $c : \mathcal{X}_d \to \mathcal{Y}$ is represented as Boolean circuit, with `False` representing $-1$ and `True` representing 1. If the output of the learner is not a valid circuit classifier, then by default we assume it classifies everything as 1. We also consider learners that can be simulated by a TM with size $\leq s$ in time $t$ using only $m$ labeled examples.

**Definition 1** (($s, m, t$)-bounded learner). *A learner $A$ is a $s$-bounded learner which outputs a classifier circuit in at most $t$ steps on any dataset consisting of at most $m$ labeled examples.*

### 2.2 Components of deep learning

In this section, we establish some notation for the building blocks of common deep learning pipelines.

**Feedforward layers.** A fully-connected feedforward layer $\mathbb{R}^{d_{\mathrm{in}}} \to \mathbb{R}^{d_{\mathrm{out}}}$, with activation function $\sigma : \mathbb{R} \to \mathbb{R}$ is parameterized by a matrix $W \in \mathbb{R}^{d_{\mathrm{out}} \times d_{\mathrm{in}}}$ and bias $b \in \mathbb{R}^{d_{\mathrm{out}}}$, specifying the map $x \mapsto \sigma(Wx+b)$, where $\sigma(\cdot)$ is applied entrywise. A feedforward network is the iterative composition of feedforward layers, possibly omitting an application of $\sigma$ at the final layer.

**Convolutional layers.** Our main construction will apply the same constant depth feedforward network repeatedly to each $3 \times 3$ patch of a 2-dimensional "image". This can be viewed as applying multi-channel convolutional layers followed by non-linear activation consecutively. Due to weight sharing across patches, the number of parameters do not depend on the input dimension but rather on the patch dimension and the number of channels. Often in practice, to ensure same output dimension as input, it is common to add a constant padding (say $p$) around the boundaries. This is crucial for our construction. More formally, a convolutional layer specifies a $k \times k$ patch-wise linear maps from $\mathbb{R}^{k \times k \times C_{\mathrm{in}}}$ to $\mathbb{R}^{C_{\mathrm{out}}}$; in particular, when $k = 1$, a convolutional layer specifies a *pixel-wise* linear map from $\mathbb{R}^{C_{\mathrm{in}}}$ to $\mathbb{R}^{C_{\mathrm{out}}}$. We let Conv2D be the application of the linear maps extended to the entire input. We overload Conv2D to also allow for patch-wise fully-connected feedforward layers.

---

[1]Natural examples where inference is more computationally expensive than learning arise in nonparametric models such as nearest-neighbors or Gaussian processes.

**Recurrent layers.** Finally, our construction will use recurrent weight sharing: for a function $f : \mathcal{Z} \times \Theta \to \mathcal{Z}$ and a positive integer $L$, we use $f^{(L)} : \mathcal{Z} \times \Theta \to \mathcal{Z}$ to denote the $L$-times iterated composition of $f$, sharing the parameters $\theta \in \Theta$ between iterations; for example,

$$f^{(3)}(X;\theta) := f(f(f(X;\theta);\theta);\theta).$$

**The training pipeline: SGD with random initialization.** Finally, we establish some notation for stochastic gradient descent, whose variants form the predominant class of methods for training neural networks. Given a continuously differentiable[2] loss function $\ell : \mathcal{Y} \times \mathcal{Y} \to \mathbb{R}$ and continuously differentiable function $f : \mathcal{X} \times \Theta \to \mathcal{Y}$ where $\Theta = \mathbb{R}^d$, a step of stochastic gradient descent (SGD) on a single example $(x,y) \in \mathcal{X} \times \mathcal{Y}$, with learning rate $\eta \in \mathbb{R}$, maps the current iterate $\theta$ to

$$\theta' := \theta - \eta \nabla_\theta \ell(f(x,\theta),y).$$

SGD on a sequence of examples $\{(x_t, y_t)\}_{t=1}^T$ is defined by applying this recurrence iteratively from an initialization $\theta_0$ (usually selected randomly from a specified distribution), giving a sequence of iterates $\{\theta_t\}_{t=1}^T$. It is routine to specify a subset $S \subseteq [d]$ of the parameters to be optimized; in this case, the parameters in $S$ are updated according to the above equation, while the rest are unchanged.

## 3 Algorithm learning and Turing-optimality

In this section, we adopt a model of learning which turns PAC learning upside down. A *learning algorithm* is a function $A$ that, for any $d, m \geq 1$, outputs a classifier $A(Z) : \mathcal{X}_d \to \mathcal{Y}$ for any dataset $Z = \{(x^{(i)}, y^{(i)})\}_{i=1}^m \in (\mathcal{X}_d \times \mathcal{Y})^m$ of $m \geq 1$ labeled $d$-dimensional examples. Recall that $\mathcal{X}_d = \{-1, 1\}^d$ and $\mathcal{Y} = \{-1, 1\}$.

The following definition captures efficient learnability of a class of learning algorithms $A$. The run-time of the algorithm is required to be polynomial in its input size $\text{poly}(dm)$. An important feature of this definition is that it requires the number of examples to be polynomial in the dimension $d$, avoiding the curse of dimensionality. Since we will soon consider $\epsilon, \delta$ as inputs, we consider only powers of 2 to avoid having to represent arbitrary real numbers.

**Definition 2** (PAO-learner). *Poly-time learning algorithm $A$ is a Probably Algorithmically Optimal (PAO) learner for family $\mathcal{A}$ if there is a polynomial $p$ such that for any $\epsilon, \delta \in 2^{-\mathbb{N}}$, for any $d \geq 1$, any distribution $\mathcal{D}$ over $\mathcal{X}_d \times \mathcal{Y}$, and any dataset sizes $m \geq 1, M \geq p(d, m, 1/\epsilon, 1/\delta)$,*

$$\Pr_{Z \sim \mathcal{D}^m, Z' \sim \mathcal{D}^M} \left[ \text{err}_\mathcal{D}(A(Z; Z')) \leq \min_{B \in \mathcal{A}} \text{err}_\mathcal{D}(B(Z)) + \epsilon \right] \geq 1 - \delta,$$

*where $Z; Z'$ is the concatenation of the two datasets $Z, Z'$. We further assume that $d, m$ and $M$ can be determined from the PAO-learner's input.*

We now observe that one can equivalently design a learning algorithm that has $\epsilon, \delta > 0$ as inputs.

**Observation 1** ($\epsilon, \delta$-PAO-learner reduction). *Let $A_{\epsilon, \delta}$ be an "$\epsilon, \delta$-PAO learner" for $\mathcal{A}$ meaning that it is a poly-time learning algorithm that takes additional inputs $\epsilon, \delta$, and there exists some constant $k$ such that: for any $\epsilon, \delta \in 2^{-\mathbb{N}}$, any dataset sizes $m \geq 1, M \geq (dm/\epsilon\delta)^k$,*

$$\Pr_{Z \sim \mathcal{D}^m, Z' \sim \mathcal{D}^M} \left[ \text{err}_\mathcal{D}(A_{\epsilon, \delta}(Z; Z')) \leq \min_{B \in \mathcal{A}} \text{err}_\mathcal{D}(B(Z)) + \epsilon \right] \geq 1 - \delta.$$

*Then, for $r = 2^{\lfloor -\frac{1}{3k} \log M \rfloor}$, $A_{r,r}$ is a PAO-learner for $\mathcal{A}$.*

The proof is straightforward and can be found in Appendix A. The $M \geq (dm/\epsilon\delta)^k$ requirement is a convenient equivalent to a polynomial bound $M \geq p(d, m, 1/\epsilon, 1/\delta)$.

Although we only analyze PAO learning for the family $\mathcal{A}$ of bounded Turing machines, it can be analyzed even for continuous classes $A$. For instance, it would be straightforward to show that grid search can PAO learn a constant number of bounded hyperparameters of a given algorithm if the

---

[2]It is routine to extend these definitions to continuous functions which are piecewise continuously differentiable, such as neural networks with ReLU activations. We omit the details in this paper, as our constructions will never evaluate a gradient of $f$ at a discontinuity.

algorithm's error is Lipschitz continuous in those hyperparameters, using a separate validation set to choose the best hyperparameters. PAO learning, as defined, does not specify how classifiers are represented, and could apply to any classifier representation. Recall that we represent classifiers by Boolean circuits as discussed in Section 2.

We next define Turing-optimal learners, which are PAO-learners for the class of bounded TMs (constant size, run in polynomial time, and output a circuit classifier).

**Definition 3** (Turing-optimal). *Fix constants $s, k \in \mathbb{N}$. Let the set $\mathcal{B}_{s,k}$ be the set of Turing machines which have $\leq s$ states and, run in time $\leq (2dm)^k$ on a dataset $Z \in (\mathcal{X}_d \times \mathcal{Y})^m$ and output a circuit. Learning algorithm $A$ is $(s, k)$-Turing-optimal if $A$ PAO-learns (or equivalently $\epsilon, \delta$-PAO learns) $\mathcal{B}_{s,k}$. Learning algorithm $A$ is Turing-optimal if $A$ is $(s, k)$-Turing-optimal for all constants $s, k \in \mathbb{N}$.*

Note that a Turing-optimal learner $A$ must run in poly-time, but the number of examples required to learn each $\mathcal{B}_{s,k}$ can be different, i.e., it will learn $\mathcal{B}_{s,k}$ using $M \geq (dm/\epsilon\delta)^{e_{sk}}$ additional examples, for a different constant $e_{sk}$ for each $s$ and $k$. Similar to Observation 1, a Turing-optimal learner can be constructed from an $(s, k)$-Turing optimal learner. The claim below, together with Observation 1, imply that a $(\epsilon, \delta, s, k)$-Turing-optimal learner can be converted to a Turing optimal learner. Algorithm 2 and its proof are presented in Appendix A.

**Claim 1** ($(s, k)$-Turing-optimal reduction). *Let $A_{s,k}$ be an algorithm that takes inputs $s, k$ and is $(s, k)$-Turing optimal for each pair of constants $s, k \in \mathbb{N}$. Then, Algorithm 2($A_{s,k}$) is Turing-optimal.*

Finally, it is not difficult to see that a Turing-optimal learner also PAC-learns any concept class $\mathcal{C}$ that is PAC-learnable. Following standard conventions, the PAC learning algorithm is given target accuracy $\epsilon$ and failure probability $\delta$ as inputs. Also, say a distribution $\mathcal{D}$ is said to be *consistent* with set $\mathcal{C}$ of classifiers if there is some $c \in \mathcal{C}$ with $\mathrm{err}_{\mathcal{D}}(c) = 0$.

**Definition 4** (PAC-learning). *Let $\mathcal{C} = \bigcup_{d \geq 1} C_d$, where $c : \mathcal{X}_d \to \mathcal{Y}$ for each $c \in \mathcal{C}_d$. oly-time[3] learning algorithm $A_{\epsilon,\delta}$ PAC-learns $\mathcal{C}$ if $A_{\epsilon,\delta}$ and there is a polynomial $p$ such that, for any $\epsilon, \delta \in 2^{-\mathbb{N}}$, $d \in \mathbb{N}$, $m \geq p(d, 1/\epsilon, 1/\delta)$ and distribution $\mathcal{D}$ consistent with $\mathcal{C}_d$:*

$$\Pr_{Z \sim \mathcal{D}^m} [\mathrm{err}_{\mathcal{D}}(A_{\epsilon,\delta}(Z)) \leq \epsilon] \geq 1 - \delta.$$

The computational polynomial-time efficiency requirement on $A_{\epsilon,\delta}$ means that its runtime is polynomial in its input size, $\mathrm{poly}(dm + \log 1/\epsilon\delta)$, because it takes $O(\log 1/\gamma)$ bits to describe $\gamma \in 2^{-\mathbb{N}}$.

**Claim 2.** *Suppose there is some learning algorithm that PAC-learns $\mathcal{C}$ and suppose that $A$ is a polynomial-time Turing-optimal learner. Then $A$ PAC-learns $\mathcal{C}$ as well.*

We defer the proof to the Appendix A.

## 4 Turing-optimality of SGD on randomly initialized RCNNs

In this section, we will design a Turing-optimal leaner in the form of a NN and a corresponding training pipeline. Our NN will be of the form of a RCNN (see Figure 2) with very few trainable parameters and our training pipeline (see Algorithm 1) will use random initialization, and random restarts to find good parameters. Let us present our main result.

**Theorem 1.** *There exists constants $c_1, c_2, c_3 > 0$ such that the following holds. For any $d, s, m, t \in \mathbb{N}$, there exists learning rate $\eta \in \mathbb{R}$, $L = c_1(t + m + d)$, and $\mathcal{U}_s \subseteq \mathbb{R}$[4] where: for any probability measure $\mathcal{D}$ on $\mathcal{X}_d \times \{-1, 1\}$, $(s, m, t)$-computable learner $A$, and training set $\mathcal{S} \in (\mathcal{X}_d \times \{-1, 1\})^m$ drawn i.i.d. from $\mathcal{D}$, Algorithm 1 returns a function $f$ such that with probability at least $s^{-c_2 s^2}$,*

$$\mathrm{err}_{\mathcal{D}}(f) \leq \mathrm{err}_{\mathcal{D}}(A, \mathcal{S}).$$

*The bit precision required by Algorithm 1 is $\lceil \log(s) \rceil + c_3$.*

---

[3]The standard PAC learning definition requires the learner to run in time also $q(d, 1/\epsilon, 1/\delta)$ for some polynomial $q$, which would admit an algorithm that is not poly-time, e.g., if it uses $m = 1$ examples but runs in time $q(d, 1/\epsilon, 1/\delta)$. However, such an algorithm can trivially be converted to a poly-time algorithm by padding its input with an additional $q(d, 1/\epsilon, 1/\delta)$ examples.

[4]This set can be constructed with knowledge of only $s$.

**Algorithm 1** SGD on randomly initialized RCNN

---

**Input:** training set $\mathcal{S} := \{(x^{(i)}, y^{(i)})\}_{i=1}^m$, size $s$, initialization set $\mathcal{U}_s$, depth $L$, learning rate $\eta$

Create dummy sample $(x^{(m+1)}, y^{(m+1)}) = (\mathbb{1}_d, 1)$

Initialize $f \in \mathcal{F}_{\mathsf{RCNN}}^{d+1, m+1, 100s, 100, L}$ (see Definition 5) with parameters $\Theta_{\mathsf{mem}}^{(1)}, \Theta_{\mathsf{rc}}, \Theta_{\mathsf{head}}$ such that $W^{(1)} = 0$, and all entries of $V_1, V_2, V_3, V_4, V_5, U_1, U_2$ are sampled uniformly from $\mathcal{U}_s$

**for** $i = 1$ **to** $m + 1$ **do**

    Update parameters in the memory layer:

$$W^{(i+1)} = W^{(i)} - \eta \, \nabla_W \ell \left( f \left( \left[ x^{(i)^\top} \; 1 \right]^\top ; W, \Theta_{\mathsf{rc}}, \Theta_{\mathsf{head}} \right), y^{(i)} \right) \Big|_{W = W^{(i)}}$$

    where $\ell : \mathbb{R} \times \mathbb{R} \to \mathbb{R}$ is the squared loss, that is, $\ell(\hat{y}, y) = \frac{1}{2}(y - \hat{y})^2$

**Output:** function $f(\cdot; W^{(m+2)}, \Theta_{\mathsf{rc}}, \Theta_{\mathsf{head}})$

---

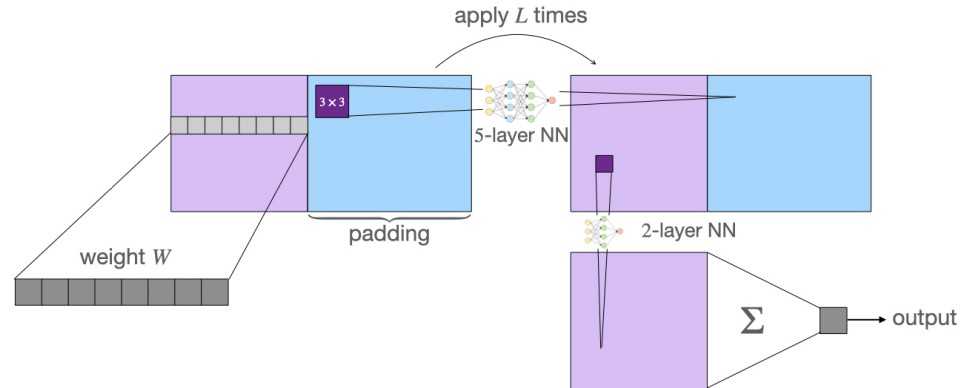

Figure 2: Recurrent convolutional neural network from our construction. A dense weight matrix $W$ is applied to the input to convert it into 2D followed by adding padding of 1s to the right. This is followed by $L$ applications of a $3 \times 3$ convolutional layer where each layer applies a 5-layer NN to each patch. This is followed by a pixel-wise convolutional layer consisting of a 2-layer NN. Lastly, the output corresponding to the main grid is summed.

**Remark.** Our Algorithm 1 sets the learning rate of 0 for the shared weights in the RCNN part of the NN and updates only the dense memory layer. Since our result is constructive, it is entirely possible that a search for an optimal-learning rate may perform better in practice.

Finally, we will use Theorem 1 to create a $(s, k)$-Turing optimal learner with the help of random restarts and an additional validation set,

**Corollary 2.** *For fixed constants $c_1, c_2 > 0$, Algorithm 1 can be converted to an $(\epsilon, \delta, s, k)$-Turing-optimal learner for any fixed $s, k$, time bound $t = (2dm)^k$ and $\epsilon, \delta \in (0, 1)$ by running it $s^{c_1 s} \log(1/\delta)$ times with random restarts, and selecting the classifier that performs best on a validation set of size $c_2 s \log s \log(1/\delta)/\epsilon^2$.*

Claim 1 converts this to a Turing-optimal learner. Because our formal definition of Turing-optimality applies only to deterministic circuit classifiers, one must also convert the NN to a circuit and derandomize the algorithm (which can be done using random bits extracted from additional iid random labeled examples (Kearns and Vazirani, 1994)).

Next we give a detailed description of the architecture and a proof overview of Theorem 1. The proof of Corollary 2 and the complete proof of Theorem 1 can be found in Appendix B.

## 4.1 Network architecture: RCNN with a memory layer

Let us first describe the neural network architecture that Algorithm 1 uses in more detail. The architecture comprises of a dense layer of size linear in $m$ (the number of samples) and $d$ (the dimension

of the input) as the first layer. The output of this layer is padded with 1s on the right, and then fed into a RCNN (recurrent weight-sharing across depth and convolutional weight sharing across width). The RCNN has only poly$(s)$ shared parameters, due to its recurrent and convolutional nature. These shared parameters are in the form of a 5-layer NN applied as a convolutional layer to $3 \times 3$ patches of the input.[5] This convolutional layer is applied recurrently $L = p(t, d, m)$ times for some polynomial $p$ (with the same parameters) where $t$ is the bound on the runtime of the TM. The final outputs from the RCNN are then passed through a pixel-wise 2-layer NN and finally summed to give a scalar prediction.

**Definition 5** (RCNN with a memory layer). *For $d_{\mathsf{in}}, d_{\mathsf{mem}}, d_{\mathsf{rc}}, d_{\mathsf{head}}, l > 0$, define the function class $\mathcal{F}_{\mathsf{RCNN}}^{d_{\mathsf{in}}, d_{\mathsf{mem}}, d_{\mathsf{rc}}, d_{\mathsf{head}}, l}$ of RCNNs where each $f \in \mathcal{F}_{\mathsf{RCNN}}^{d_{\mathsf{in}}, d_{\mathsf{mem}}, d_{\mathsf{rc}}, d_{\mathsf{head}}, l}$ is parameterized by memory layer parameters $\Theta_{\mathsf{mem}} = \{W\}$ with $W \in \mathbb{R}^{d_{\mathsf{mem}} \times d_{\mathsf{in}}}$, RC layer parameters $\Theta_{\mathsf{rc}} := \{V_1, V_2, V_3, V_4, V_5\}$[6] with $V_1 \in \mathbb{R}^{d_{\mathsf{rc}} \times 9}$, $V_2, V_3, V_4 \in \mathbb{R}^{d_{\mathsf{rc}} \times d_{\mathsf{rc}}}$, $V_5 \in \mathbb{R}^{1 \times d_{\mathsf{rc}}}$, and head layer parameters $\Theta_{\mathsf{head}} = \{U_1, U_2\}$ with $U_1 \in \mathbb{R}^{d_{\mathsf{head}} \times 1}, U_2 \in \mathbb{R}^{1 \times d_{\mathsf{head}}}$). The local convolutional operation is denoted by $f_{\mathsf{Conv2D}} : \mathbb{R}^{3 \times 3} \to \mathbb{R}$ and is applied on the $3 \times 3$ grid centered at each coordinate of the $d_{\mathsf{mem}} \times (d_{\mathsf{in}} + l)$ input (with padding $p$ for the edge coordinates). We overload the notation $f_{\mathsf{Conv2D}}$ to denote the $d_{\mathsf{mem}} \times (d_{\mathsf{in}} + l)$ output post the local application on the input.*

$$f(z; \Theta_{\mathsf{mem}}, \Theta_{\mathsf{RCNN}}; \Theta_{\mathsf{head}}) = \mathsf{sum}\left(f_{\mathsf{head}}\left(f_{\mathsf{Conv2D}}^{(l)}\left(f_{\mathsf{mem}}(z; \Theta_{\mathsf{mem}}); \Theta_{\mathsf{RCNN}}\right)_{:,:d_{\mathsf{in}}}\right); \Theta_{\mathsf{head}}\right)$$

$$\text{where} \ \ f_{\mathsf{mem}}(z; \Theta_{\mathsf{mem}} = \{W\}) = [\ W \mathrm{diag}(z) \quad \mathbb{1}_{d_{\mathsf{mem}} \times l}\ ]$$
$$f_{\mathsf{head}}(z : \theta = \{U_1, U_2\}) = U_1 \sigma(U_2 z)$$
$$f_{\mathsf{Conv2D}}(z; \Theta = \{V_1, V_2, V_3, V_4, V_5\}) = V_5 \sigma(V_4 V_3 \sigma(V_2 V_1 \mathsf{vec}(z))))).$$

*with $\sigma$ being the ReLU activation.*

**Remark.** If our architecture did not have both recurrent and convolutional weight sharing, then the number of parameters would have dependence on $d_{\mathsf{in}}, d_{\mathsf{mem}}$ and $l$, which depend on $m, d$, and $T$.

## 4.2 Proof sketch for Theorem 1

Here we present a proof sketch for Theorem 1. Our proof follows by construction, that is, we show that for each TM $A$ of size $s$, there exists parameters $\Theta_{\mathsf{rc}}$ and $\Theta_{\mathsf{head}}$ that ensure that (1) for the first $m + 1$ steps, when $\Theta_{\mathsf{mem}}$ is trained with SGD, the gradients assist with memorizing the training set in the values of $W_{\mathsf{mem}}$, and (2) given the memorized training set, the RCNN computes the roll-out of $A$ with the input tape having the training set and the test example giving the prediction of $A$ on the test example as the output. We finally show that the parameters $\Theta_{\mathsf{rc}}$ and $\Theta_{\mathsf{head}}$ in our construction for all TM $A$ of fixed size and runtime belong to a fixed finite set of size $O(s)$ that can be constructed with knowledge of only $s$. The following lemma summarizes the aforementioned properties:

**Lemma 1.** *For any $d, s, m, t \in \mathbb{N}$, $\delta \in (0, 1)$, and any $(s, m, t)$-computable learner $A$, there exists $\Theta_{\mathsf{rc}}, \Theta_{\mathsf{head}}$ with each parameter belonging to a fixed set $\mathcal{U}_s$ of size $O(s)$ (that can be constructed with the knowledge of only $s$) such that Algorithm 1 with $L =$ satisfies:*

*1. Memorization: For $1 \leq i \leq m + 1$, $W_{ab}^{(i)} = \begin{cases} \frac{1}{3} y^{(a)} x_b^{(a)} & \text{if } a \leq i, b \leq d \\ \frac{1}{3} y^{(a)} & \text{if } a \leq i, b = d + 1 \\ 0 & \text{otherwise.} \end{cases}$*

*2. Computation: For all $x \in \{\pm 1\}^d$, $f\left(\begin{bmatrix} x \\ 1 \end{bmatrix}; W^{(m+2)}, \Theta_{\mathsf{rc}}, \Theta_{\mathsf{head}}\right) = A(\mathcal{S})[x]$.*

Theorem 1 follows from the above the computation property of Lemma 1, since it implies that the error of Algorithm 1 will be exactly equal to the error of $A$.

What remains is to prove the existence of parameters that satisfy Lemma 1. Let us now briefly describe the key functionality we require the RCNN to implement for this:

**Computation.** Each roll-out step of the TM is a local update around the head of the TM. To implement one step of the TM, we need to compute the transition function of $A$ at the location of the

---

[5]Note that we did not optimize our construction. It is quite possible to improve this to a shallower network.

[6]For ease of presentation, we hide the biases from the parameters. We can assume that the input is padded with 1 to account for biases.

head, update the new head and state, and copy the inputs of the rest of the tape. Our first observation is that this local update can be implemented using a convolutional layer if we interpret the input as the tape of $A$ (input and working concatenated) with the head and state information stored along with the tape value. Composing these layers $t$ times (with the same parameters) allows us to simulate $t$ steps of the TM. More importantly, the convolutional layer requires only $O(s)$ parameters since $A$ only has $s$ states. In order to decode the tape content, head position, and state from the values fed to the layers of the RCNN, we interpret the input in base 3 and use different positions to encode the desired information (see Appendix B.1 for more details).

**Memorization.** Given that we can simulate the TM, we need to ensure that the input to the RCNN has the training set and the test example encoded onto it. Similar to Abbe and Sandon (2020), we can use SGD to memorize the training examples into the weights of the memory layer, with each row storing one example. We do this by ensuring that the gradient at iteration $i$ through the RCNN is 1 for exactly the $i$th row and 0 for every other row. We also ensure that the output is 0 through the memorization phase. By using chain rule, this gives us a gradient of $[y^{(i)}x^{(i)}, y^{(i)}]$ for the $i$th row of $W_{\mathsf{mem}}$ and 0 otherwise. We can this in a local manner using the RCNN. Note that the memory layer has $O(dm)$ parameters, however we can learn these parameters from 0 initialization.

**Communication.** Lastly, we need to ensure that the network can differentiate between memorization phase (passing meaningful gradients) and computation phase (implementing the roll-out of the TM) using the local operations in the RCNN layers. We do this by implementing a local communication protocol: we broadcast a message based on certain conditions, where each RCNN layer implements a step of the broadcast. In order to broadcast to the entire input, we require $\approx d + m$ overhead in terms of the depth of the network.

Finally, we show that the above mentioned functionality can be achieved by a 5-layer NN with $O(s^2)$ parameters. To do so, we first describe the exact function we require the network to compute and its Jacobian on all inputs that our bit precision allows (see Appendix B.2 for the exact function). Given the function and its Jacobian on a finite set, we prove a general representational theorem (see Appendix B.4.1) that constructs a 5-layer NN with weights from a fixed set that can be constructed with knowledge of only $s$. We refer the reader to Appendix B for the complete proof.

## 5   Discussion

In this section, we discuss potential practical implications of Turing-optimality, and broadly discuss corroborations and tensions with empirical trends.

**Discovering reusable algorithms.** Our analysis is wasteful in that, if one has multiple learning problems, e.g., multiple parity problems, one has to relearn the learning algorithm for each one. In fact, arguably the NN may not have learned an algorithm for parity problems in general, but rather a specialized algorithm that works on just one. To find a *reusable* algorithm, one needs multiple problems, say drawn from a meta-distribution $\mu$ over learning problems. The idea is simple: viewing the constant number of weights of the RCNN filter as hyperparameters, one tries multiple such hyperparameters on $\log 1/\delta$ learning problems, and finally selecting the hyperparameters that perform best on average. With a constant-sized random sample of hyperparameters, with high probability, one of them will perform nearly as well as the best constant-sized TM not only on these few training problems, but also on future problems drawn from $\mu$. This is the setting considered by the literature on meta-learning (Hospedales et al., 2021) and data-driven algorithm design (Balcan, 2020). Recently, Garg et al. (2022) show empirically that Transformer architectures can meta-learn and execute simple learning algorithms in-context. We leave this and numerous other interesting directions for future work.

**Concise architectures.** Many Turing-complete architectures have been proposed and used in practice. The lens of Turing-optimality may help us understand what architectures are minimally adequate from a theoretical perspective. In particular, it has been popular to report ever-growing parameter counts for state-of-the-art models in domains such as natural language processing (Brown et al., 2020; Fedus et al., 2021; Lin et al., 2021). Although the other benefits of over-parameterization are at play, this work suggests that very parameter-efficient architectures are sufficient to simulate any computationally efficient learning algorithm. In light of the above, one concrete direction for further investigation is to develop practical variants of our RCNN construction, in domains dominated by other architectures. Although our analysis is too pessimistic to be of immediate practical use, it

highlights the computational power of an architecture that has occasionally appeared in applications-focused research (Pinheiro and Collobert, 2014; Liang and Hu, 2015; Spoerer et al., 2017; Alom et al., 2021). Significantly closer to our work, Schwarzschild et al. (2021) conduct an empirical study on the ability of RCNNs to extrapolate from easier to harder tasks (thus "learn an algorithm"); our work shows that it is possible for these architectures to learn any computationally efficient algorithm. Similarly, RCNNs have been investigated for planning in RL (Tamar et al., 2016); other empirical works which take a "computation time" view of depth include (Graves et al., 2014; Banino et al., 2021; Kaiser and Sutskever, 2015). RCNNs have not seen widespread adoption in state-of-the-art deep learning compared to their non-recurrent and/or non-convolutional counterparts.[7]

**Beyond local search for recurrent models.** The proliferation of non-recurrent attention-based models in domains previously dominated by recurrent networks, along with the under-representation of RCNNs, is perhaps due to instabilities in training recurrent networks with SGD (Pascanu et al., 2013). Indeed, Kasai et al. (2021) demonstrate that a carefully designed training procedure can convert a trained Transformer into a more parameter-efficient RNN. There may be undiscovered practical training algorithms which can bridge the gap in favor of recurrent models. Although our Turing-complete algorithm uses SGD, it uses the gradients in a way that is far from making local greedy progress on an objective; the as-efficient-as-possible search for the correct TM is implemented by random initialization.

**Exhaustive search.** Our Turing-optimal training pipeline relies upon exhaustively comparing classifiers trained from different random initializations to choose the best classifier within the class of concise Turing machines. This runs counter to the classical viewpoint from continuous optimization, where gradient descent is seen as a *local search* method. In problem settings of a combinatorial or algorithmic nature, we posit that exhaustive search may be unavoidable; indeed, state-of-the-art pipelines already include forms of exhaustive search such as beam search and its variants (Reddy et al., 1977; Leblond et al., 2021), as well as chain-of-thought generation (Wang et al., 2022).

**Memory modules.** There have been many attempts to build practical memory modules into neural networks (Graves et al., 2014; Sukhbaatar et al., 2015; Grave et al., 2016; Dai et al., 2019). Our construction proposes an integrated memory mechanism: use SGD to store samples in the first layer's trainable parameters, ahead of the deep (RCNN) computation layers, by carefully ensuring that the gradient signal back-propagates through the RCNN layers correctly.

# 6 Conclusion

In this paper, we present a simple NN architecture, combining recurrent and convolutional weight sharing, that achieves Turing-optimality. Among other things, it learns the well-studied class of parity functions in polynomial time, whereas prior NN analyses of parity require time exponential in the size of the parity function (or require a parity learning algorithm to be initialized into the networks weights). Our proposed architecture has connections to the deep learning literature and observed empirical trends, discussed in Section 5. Immediate improvements to make the architecture more concise and natural include: (1) reducing the size of the dense parameters to depend on the algorithm's memory usage instead of the training sample size, and (2) using SGD beyond memorization. In future work, it would be interesting to understand which other architectures are Turing-optimal, answering questions such as: are 2D convolutions necessary, and are there natural Transformer-based training pipelines which are Turing-optimal?

**Limitations and broader impact.** The primary limitation of this work is that the constant factors in our analysis are much too large to be meaningful in practice. Nonetheless, we hope that idea of combining recurrent and convolutional weight sharing will have impact. Also, the algorithms found using enumerative program search would be, by default, difficult to interpret. Using such algorithms carries risks, especially if the algorithm is not doing what one expects it to do.

**Acknowledgments.** We thank Santosh Vempala for useful discussions. Sham Kakade acknowledges funding from the Office of Naval Research under award N00014-22-1-2377 and the National Science Foundation Grant under award #CCF-1703574.

---

[7]Note that the popular R-CNN of (Girshick et al., 2014) is not recurrent; the "R" stands for *region*.

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
