---
**Algorithm 2** Reduce $(s, k)$-Turing-optimal learning to Turing-optimal learning.

---
**Input:** learner $A_{s,k}$ (that takes inputs $s, k$), $d, m, M \in \mathbb{N}$, $Z \in (\mathcal{X}_d \times \mathcal{Y})^m$, $Z' \in (\mathcal{X}_d \times \mathcal{Y})^M$
Let $Z'[: t]$ denote the first $t$ examples of $Z'$ and $Z'[-r :]$ denote the last $r$ examples
$r = M^{1/10}$
**for** $s = 1$ **to** $r$ **do**
   **for** $k = 1$ **to** $r$ **do**
      **for** $t = 1$ **to** $r$ **do**
         Let $c_{s,k,t} = A_{s,k}(Z; Z'[: t])$, running $A_{s,k}$ with a maximum time limit of $M$.
         Let $\hat{e}_{s,k,t}$ be the empirical error of $c_{s,k,t}$ over $Z'[-r :]$
**Output:** $c_{s,k,t}$ for $s, k, t$ that minimize $\hat{e}_{s,k,t}$.

---

# A   Turing-optimality: Proofs of Observation 1 and Claims 1 and 2

We first restate and prove Observation 1, that an $\epsilon, \delta$-PAO learner $A_{\epsilon,\delta}$ that requires $M \geq (dm/\epsilon\delta)^k$ examples gives rise to a PAO-learner $A_{r,r}$ for $r = 2^{\lfloor -\frac{1}{3k} \log M \rfloor}$.

**Observation 1** ($\epsilon, \delta$-PAO-learner reduction). *Let $A_{\epsilon,\delta}$ be an "$\epsilon, \delta$-PAO learner" for $\mathcal{A}$ meaning that it is a poly-time learning algorithm that takes additional inputs $\epsilon, \delta$, and there exists some constant $k$ such that: for any $\epsilon, \delta \in 2^{-\mathbb{N}}$, any dataset sizes $m \geq 1$, $M \geq (dm/\epsilon\delta)^k$,*

$$\Pr_{Z \sim \mathcal{D}^m, Z' \sim \mathcal{D}^M} \left[ \mathrm{err}_{\mathcal{D}}(A_{\epsilon,\delta}(Z; Z')) \leq \min_{B \in \mathcal{A}} \mathrm{err}_{\mathcal{D}}(B(Z)) + \epsilon \right] \geq 1 - \delta.$$

*Then, for $r = 2^{\lfloor -\frac{1}{3k} \log M \rfloor}$, $A_{r,r}$ is a PAO-learner for $\mathcal{A}$.*

Note that $r$ is defined so as to be a power of 2 in $[M^{-1/3k}/2, M^{-1/3k}]$.

*Proof.* Let $p(d, m, \epsilon, \delta) := (\epsilon\delta)^{-3k} + (4dm)^{3k}$. For any $M \geq p(d, m, \epsilon, \delta)$, note first that, by definition of $r$, $M^{-1/3k}/2 \leq r \leq M^{-1/3k} \leq \epsilon\delta \leq \max(\epsilon, \delta)$. We also claim that $M \geq (dm/r^2)^k$. To see this, note that,

$$(dm/r^2)^k \leq (4dmM^{2/3k})^k = (4dm)^k M^{2/3} \leq M,$$

for our polynomial $p$. Thus $A_{r,r}$ satisfies the requirements of a PAO-learner, since $r \leq \epsilon$ and $r \leq \delta$. $\qquad\square$

The reduction used in Claim 1 to show the equivalence of Turing-optimal and $(s, k)$-Turing-optimal, is shown in Algorithm 2. Note that the algorithm was chosen for its simplicity rather than optimizing parameters.

**Claim 1** ($(s, k)$-Turing-optimal reduction). *Let $A_{s,k}$ be an algorithm that takes inputs $s, k$ and is $(s, k)$-Turing optimal for each pair of constants $s, k \in \mathbb{N}$. Then, Algorithm 2$(A_{s,k})$ is Turing-optimal.*

*Proof.* Since $A_{s,k}$ is a PAO learner for $\mathcal{B}_{s,k}$, there is some polynomial $p(d, 1/\epsilon, 1/\delta)$ such that, for any $\epsilon, \delta \in 2^{-\mathbb{N}}$, given $M \geq p(d, m, 1/\epsilon, 1/\delta)$ additional examples, with probability $\geq 1 - \delta/2$, it outputs a classifier with error within $\epsilon/2$ of the best in $\mathcal{B}_{s,k}$. Also, $A_{s,k}$ runs in time in $q(d, m + M)$ for some polynomial $q$. We must show that Algorithm 2 is also a PAO learner for $\mathcal{B}_{s,k}$, even though it does not take $s, k$ as inputs. Let $M' = p(d, m, 2/\epsilon, 2/\delta)$. As long as $r \geq \max(s, k, M')$, and as long as $M \geq q(d, m + M')$, we will consider $c_{s,k,t}$ as one of the candidate classifiers. Both will be the case for,

$$M \geq \left(s + k + p(d, m, 2/\epsilon, 2/\delta)\right)^{10} + q\left(d, m + p(d, m, 2/\epsilon, 2/\delta)\right) = \mathrm{poly}(d, m, 1/\epsilon, 1/\delta).$$

Finally, clearly Algorithm 2 runs in polynomial time, i.e., time $\mathrm{poly}(d, m + M)$ due to the timeout and number of iterations. $\qquad\square$

We now move to the poof of Claim 2. Note that the Turing-optimal learner $A$ is a PAC learner though $A$ does not even require $\epsilon, \delta$ as inputs.

**Claim 2.** *Suppose there is some learning algorithm that PAC-learns $\mathcal{C}$ and suppose that $A$ is a polynomial-time Turing-optimal learner. Then $A$ PAC-learns $\mathcal{C}$ as well.*

*Proof.* Call the PAC learner $P_{\epsilon,\delta}$. Let constant $k$ be such that both $P_{\epsilon,\delta}$ runs in time $\leq (dm + \log(1/\epsilon\delta))^k$ and uses $p(d, 1/\epsilon, 1/\delta) \leq (d/\epsilon\delta)^k$ examples. In particular, for $r = m^{-1/3k}$, $P_{r,r}$ is a learning algorithm that, when it is given $m \geq (d/r^2) = d^k m^{2/3k}$ (equivalently $m \geq d^{3k}$) examples, with probability $\geq 1 - r$ outputs a classifier with error at most $r$. And $P_{r,r}$ runs in time at most $(dm + \log(m))^k \leq (2dm)^k$. Thus as long as $m \geq (8/\min(\epsilon, \delta)^3)^k$, with probability $\geq 1 - \delta/2$ it outputs a classifier with error at most $\epsilon/2$. Since $A$ is a Turing-optimal learner, it outputs a classifier whose error is within $\epsilon/2$ of $P_{r,r}$, with probability $\geq 1 - \delta/2$, using additional samples $M = \text{poly}(d, m)$. By the union bound, this means that with probability $\geq 1 - \delta$, it outputs a classifier with error at most $\epsilon$ as required by PAC learning. $\qquad\square$

# B  Proof of Theorem 1

In this section we will give a complete proof of 1. We will keep this self contained and repeat necessary content from the main paper.

As discussed before, our proof follows by construction, that is, we show that for each TM $A$ of size $s$, there exists parameters $\Theta_{\text{rc}}$ and $\Theta_{\text{head}}$, such that, when Algorithm 1 is run with these parameters:

- *Memorization*: for the first $m + 1$ steps, when $\Theta_{\text{mem}}$ is trained with SGD, the gradients assist with memorizing the training set in the values of $W$

- *Computation*: given the memorized training set, the RCNN computes the roll-out of $A$ with the input tape having the training set and the test example giving the prediction of $A$ on the test example as the output.

We then show that we can choose the parameters $\Theta_{\text{rc}}$ and $\Theta_{\text{head}}$ in our construction such that for all TM $A$ of size $s$ belong to a fixed finite set of size $O(s)$ that can be constructed with knowledge of only $s$.

Let us restate Lemma 1 (more formally):

**Lemma 2** (Restatement of Lemma 1). *There exists constants $c'_1, c'_2, c'_3$ such that: for any $d, s, m, t \in \mathbb{N}$, there exists a fixed initialization set $\mathcal{U}_s$ of size $c'_1 s$ where: for any $(s, m, t)$-computable learner $A$, there exists $\Theta_{\text{rc}}, \Theta_{\text{head}}$ such that for all $\mathcal{S} \in (\mathcal{X}_d, \mathcal{Y})^m$, Algorithm 1 run on $\mathcal{S}$ with $l = c'_2(t + m + d)$, $\eta = 1/54$, and initialization set $\mathcal{U}_s$, satisfies:*

*1. Memorization: For $1 \leq i \leq m + 1$, $W_{ab}^{(i)} = \begin{cases} 3^{-1} y^{(a)} x_b^{(a)} & \text{if } a < i, b \leq d \\ 3^{-1} y^{(a)} & \text{if } a < i, b = d + 1. \\ 0 & \text{otherwise.} \end{cases}$*

*2. Computation: For all $x \in \{\pm 1\}^d$, $f\left( \begin{bmatrix} x \\ 1 \end{bmatrix}; W^{(m+2)}, \Theta_{\text{rc}}, \Theta_{\text{head}} \right) = A(\mathcal{S})[x]$.*

Using the above lemma (part 2), we know that there exists parameters $\Theta_{\text{rc}}, \Theta_{\text{head}}$ that allow for our training pipeline to output exactly the function (say $f$) that $A$ learns on training set $\mathcal{S}$. Note that our construction implements the underlying learning algorithm $A$ and is able to provide this guarantee for all training sets $\mathcal{S}$ simultaneously. Thus, this implies that $\text{err}_{\mathcal{D}}(f) = \text{err}_{\mathcal{D}}(A, \mathcal{S})$ for any distribution $\mathcal{D}$. The chance that random initialization in Algorithm 1 will find these parameters is at least $s^{-cs^2}$ for fixed some $c > 0$ since there are $\approx 10^4 s^2$ parameters and each parameter has $c'_1 s$ potential values. This proves Theorem 1.

**Organization**  . In the remaining section we will prove Lemma 2. We will first describe how our construction will interpret the data values in $O(\log(s))$ precision (B.1), and the type of TMs we will consider. We will then describe the exact function of the RCNN layer and its corresponding Jacobian on the inputs (B.2). We will then show how SGD uses this functionality to memorize the training

examples (B.3). Finally, we show that the desired functionality of the RCNN layer can be achieved by a 5-layer NN with $O(s^2)$ parameters (B.4). [8]

## B.1 Data representation and TM modifications

### B.1.1 Data

In order to implement the required functionality of the RCNN layers, we will need $O(\log s)$ bit precision for each input to store relevant information. We work in base 3 to allow for storing $\{-1, 0, 1\}$ uniquely in each bit. More formally, the representation of each input/output in the network will have bit precision $\log(s) + 4$. Our construction will ensure that the inputs/outputs at each point are exactly representable with this bit precision in the following form: each input/output $a = \sum_{i=0}^{\log_2(s)+4} a(i)3^{-i}$ where $a(i) \in \{-1, 0, 1\}$ for $i \in \{0, 1, \ldots, \log_2(s) + 4\}$. The indices will encode different functionalities as follows:

- *Index 0* indicates whether the input is part of the padding (1) or not (0).
- *Index 1* indicates the content on the tape: blank symbol (0) or $\pm 1$.
- *Index 2* indicates whether the current phase in the training is memorization (-1), computation (1), or unknown (0).
- *Index 3* indicates the presence of the head: 1 implies head is present, and 0 indicates no head.
- *Index* $4, \ldots, \log_2(s) + 4$ indicate the state of the TM $\{0, 1\}^{\log(s)}$, and are only non-zero when the head is set, that is, bit $3 \neq 0$.

For ease of presentation, let us define $\mathsf{val} : \mathbb{R} \times \mathbb{Z}_{\geq 0} \to \{-1, 0, 1\}$ as the function that given input $a$ and index $i$, extracts bit $i$, that is, $a(i)$ if $a = \sum_{j=0}^{\log(s)+4} a(j)3^{-j}$ and is undefined otherwise. We also define $\mathsf{state} : \mathbb{R} \to \mathbb{R}$ which given input $a$ and index $i$, extracts the value corresponding to the state, that is $\sum_{j=4}^{\log(s)+4} a(j)3^{-j+4}$. Let us define the set of possible values satisfying the above by $\mathcal{V}_s := \left\{ \sum_{i=1}^{\log(s)+4} a(i)3^{-i} : a(i) \in \{-1, 0, 1\} \right\}$.

### B.1.2 Turning Machines (TMs)

Let us formally define one-tape TMs,

**Definition 6** (Turing machine). *A Turing machine is defined as a tuple $\langle Q, \Gamma, \delta, F \rangle$ where $\Gamma$ is a finite non-empty set of tape alphabet symbols, $Q$ is a finite set of states, $F \subseteq Q$ is the final state indicating accept or reject, and $\delta : Q \times \Gamma \to Q \times \Gamma \times \{0, 1\}$ is the transition function where 0, 1 are left and right shifts. Given input $x \in \Gamma^*$ put in the start of the tape, let $\mathcal{TM}_{\langle Q, \Gamma, \delta, F \rangle}(x) \in \{0, 1\}$ denote the output the TM produces if it halts on $x$. We denote the size of the TM $s$ by the number of states, that is, $s = |Q|$.*

For technical ease, our algorithm will be competitive with all TMs of size $s$ with the following modifications:

- *States*: We do a 1-1 mapping of state space $Q$ of size $s$ to $\{\sum_{i=4}^{\log(s)+4} a_i 3^{-i} | a_i \in \{-1, 1\}\}$ with 0 being the start state.
- *Single tape*: Instead of assuming 2 tapes with the training set on one tape and the test example on another, we will assume that they are concatenated onto one tape, followed by the working tape.
- *2D tape instead of 1D tape*: TM will run on a 2D tape with functionality for up, down, left, right, and no move. This implies that the transition function will have the following form $\delta : Q \times \Gamma \to Q \times \Gamma \times \{\uparrow, \downarrow, \leftarrow, \rightarrow, \times\}$. The 2D tape allows us to have markers for the number of samples and data dimension without requiring that in the state space.
- *First step*: We assume that the machine's first step is to not move and not change the tape symbol. This is useful for starting the communication protocol.

---

[8]In the main submission, we erroneously wrote $s$ instead of $s^2$ parameters. We have corrected this in the current version.

- *XOR input*: In order to compute $A(\mathcal{S})[x]$ for $\mathcal{S} = \{(x^{(i)}, y^{(i)})_{i=1}^m\}$, we will have a 2D matrix of size $(m+1) \times (d+1)$ with the $m+1$th row containing $[x^\top \ 1]$, and row $i \ (\leq m)$ containing $y^{(i)} \left[ (x \circ x^{(i)})^\top \ y^{(i)} \right]$ where $\circ$ is the coordinate-wise dot product.

- *Halting position*: TM halts with output on the left-top corner of the tape and the rest of the input tape set to blank. Working tape can have any value.

Standard reductions show that two tape TMs can be implemented using 1 tape TMs. It is not hard to see that the TM with the 2D tape can implement any TM on the 1D tape. Not moving on the first step can be made possible by adding a single additional state with a path to the starting state. As for the XOR input tape, this can be converted to the original tape by adding extra states to multiply each coordinate of $x$ down the column and multiplying $y^{(i)}$ across the row. Lastly, we can use two additional states to make the output 0 on the entire input tape except the left-top corner with the output. All these conversion causes a blow up of a constant number in states and polynomial in $m, d$ extra runtime. We skip the details here as these follow from standard reductions of TMs, and assume from now on that our TM has the above form.

## B.2 RCNN functionality

Here we will describe the exact function the RCNN layer ($f_{\mathsf{Conv2D}}$) implements and its Jacobian. We will describe how to convert this into a 5-layer NN in the subsequent sections.

Our RCNN layers take in $3 \times 3$ grid around each input coordinate with the coordinate as the center. Our construction ensures that each input/output coordinate has representation as above. For the corner coordinates, we consider padding $p = 1/6$ so it is outside our bit representation. This allows it to be identified distinctly from any value of the input/output.

Now, we would like for all $X = \begin{pmatrix} x_{-1,-1} & x_{-1,0} & x_{-1,1} \\ x_{0,-1} & x_{0,0} & x_{0,1} \\ x_{1,-1} & x_{1,0} & x_{1,1} \end{pmatrix}$ such that each entry is in the set $\mathcal{V}_s$,

the function computed by $\bar{f}_{\mathsf{Conv2D}}$ has three modes depending on the value of index 2 corresponding to the phase: *message passing and memorization* ($\bar{f}_{\mathsf{msg}}$), *computation* ($\bar{f}_{\mathrm{TM}}$), and *identity pass through*.

$$\bar{f}_{\mathsf{Conv2D}}(X) = \begin{cases} \bar{f}_{\mathsf{msg}}(X) & \text{if } \underbrace{\mathsf{val}(x_{0,0}, 2) = 0}_{\text{is phase unknown?}}, \\ \bar{f}_{\mathrm{TM}}(X) & \text{if } \underbrace{\mathsf{val}(x_{0,0}, 2) = 1}_{\text{is compute phase?}}, \\ x_{0,0} & \text{otherwise.} \end{cases}$$

Here, $\bar{f}_{\mathsf{msg}}$ runs the message passing protocol which identifies the correct phase and then broadcasts it to all inputs. It also assists with memorization, by identifying the location to memorize and ensuring that the gradients are zero for all non-memorizing coordinates. Let us formally describe the *message passing and memorization* functionality:

$$\bar{f}_{\mathsf{msg}}(X) = \begin{cases} \underbrace{-3^{-2}}_{\text{memory phase}} & \text{if } \underbrace{x_{0,0} = 0}_{\text{not memorized?}} \text{ and } \underbrace{(x_{-1,0} = p \text{ or } \mathsf{val}(x_{-1,0}, 1) \neq 0)}_{\text{is first row or is above row memorized?}} \\ \underbrace{-3^{-2}}_{\text{memory phase}} & \text{if } \underbrace{\mathsf{val}(x_{-1,0}, 2) = -1 \text{ or } \mathsf{val}(x_{1,0}, 2) = -1}_{\text{are below or above coordinates in memory phase?}} \\ x_{0,0} + \underbrace{3^{-2}}_{\text{compute phase}} + \underbrace{3^{-3}}_{\text{head}} & \text{if } \underbrace{\mathsf{val}(x_{0,0}, 1) \neq 0 \text{ and } x_{0,1} = 1 \text{ and } x_{1,0} = p}_{\text{is row memorized and is coordinate bottom-left coordinate?}} \\ x_{0,0} + \underbrace{3^{-2}}_{\text{compute phase}} & \text{if } \underbrace{\mathsf{val}(x_{-1,0}, 2) = 1 \text{ or } \mathsf{val}(x_{1,0}, 2) = 1 \text{ or } \mathsf{val}(x_{0,-1}, 2) = 1 \text{ or } \mathsf{val}(x_{0,1}, 2) = 1}_{\text{are any of the neighbouring coordinates in compute phase?}} \\ x_{0,0} & \text{otherwise.} \end{cases}$$

The invariant that is maintained during training is that samples are memorized row wise in the weights of $W$. Since the weights are 0 for rows that are yet to be memorized, we can identify the row to memorize (first if condition). Similarly, once all rows are memorized, it can be identified at the last row and computation phase can begin with the head being assigned (third if condition). See Figure

for a visual explanation of this. Post identification, our message passing protocol essentially lets the other entries set the phase themselves (second and fourth if conditions). In the memorization phase, this is supplemented with zeroing the output (and the gradient) of any irrelevant coordinate.

Now we are ready to describe the *TM Roll-out Functionality*. Once the compute phase is established and the head is assigned. Observe that our TM does not move in the first step in order to ensure that the compute phase message is broadcast one step ahead of the movement of the TM.

$$
\bar{f}_{\text{TM}}(X) = \begin{cases}
\underbrace{\gamma(x_{0,0})[1]/3}_{\text{new tape value}} + \underbrace{3^{-2}}_{\text{compute phase}} + \underbrace{\mathbb{1}[\gamma(x_{0,0})[2] = \times](3^{-3} + \gamma(x_{0,0})[0]3^{-4})}_{\text{update head and state if it is not moving}} & \text{if } \underbrace{\mathsf{val}(x_{0,0}, 3) = 1}_{\text{is head?}} \\
x_{0,0} + \underbrace{\mathbb{1}[\gamma(x_{1,0})[2] = \uparrow](3^{-3} + \gamma(x_{1,0})[0]3^{-4})}_{\text{update head and state if it is moving to the current coordinate}} & \text{if } \underbrace{\mathsf{val}(x_{1,0}, 3) = 1}_{\text{is head below?}} \\
x_{0,0} + \underbrace{\mathbb{1}[\gamma(x_{-1,0})[2] = \downarrow](3^{-3} + \gamma(x_{-1,0})[0]3^{-4})}_{\text{update head and state if it is moving to the current coordinate}} & \text{if } \underbrace{\mathsf{val}(x_{-1,0}, 3) = 1}_{\text{is head above?}} \\
x_{0,0} + \underbrace{\mathbb{1}[\gamma(x_{0,1})[2] = \leftarrow](3^{-3} + \gamma(x_{-1,0})[0]3^{-4})}_{\text{update head and state if it is moving to the current coordinate}} & \text{if } \underbrace{\mathsf{val}(x_{0,1}, 3) = 1}_{\text{is head on the right?}} \\
x_{0,0} + \underbrace{\mathbb{1}[\gamma(x_{0,-1})[2] = \rightarrow](3^{-3} + \gamma(x_{0,-1})[0]3^{-4})}_{\text{update head and state if it is moving to the current coordinate}} & \text{if } \underbrace{\mathsf{val}(x_{0,-1}, 3) = 1}_{\text{is head on the left?}} \\
x_{0,0} & \text{otherwise.}
\end{cases}
$$

for $\gamma(x) = \delta(\mathsf{state}(x), \mathsf{val}(x, 1))$ is the output of the transition matrix for the current state, head, and tape contents. Here index 0 is the new state, index 1 is the new tape contents, and index 2 is the direction of head movement. In the above functionality, the checks identify where the head is and update the TM based on the current state and tape contents. For the coordinates not adjacent to the head, the function just passes through the tape value. This implements one step of the TM.

Finally, in order to ensure that gradients are passed through during memorization, we require the gradients to satisfy,

$$
\nabla_{x_{i,j}} \bar{f}_{\text{Conv2D}}(X) = \begin{cases}
1 & \text{if } i = j = 0 \text{ and } \underbrace{\mathsf{val}(x_{0,0}, 2) = -1}_{\text{is memory phase?}} \\
1 & \text{if } i = j = 0 \text{ and } \underbrace{\mathsf{val}(x_{0,0}, 2) = 0}_{\text{is phase unknown?}} \text{ and } \underbrace{x_{0,0} = 0}_{\text{not memorized?}} \text{ and } \underbrace{(x_{-1,0} = p \text{ or } \mathsf{val}(x_{-1,0}, 1) \neq 0)}_{\text{is first row or is above row memorized?}} \\
0 & \text{otherwise.}
\end{cases}
$$

Observe that, the above formulation act as a filter, to allow gradients to pass through only when the convolution starts the memorization, and as long as the memorization continues. This allows us to pass gradients down to only the row of $W$ that is memorizing the example. We will explain this in more detail in the next section.

Finally, we want $\bar{f}_{\text{head}} : \mathbb{R} \to \mathbb{R}$ to implement the following function

$$
\bar{f}_{\text{head}}(x) = \begin{cases}
-1 & \text{if } x \leq -1/6 \\
18(x + 3^{-2}) & \text{if } x \in (-1/6, 0) \\
-18(x - 3^{-2}) & \text{if } x \in (0, 3^{-2}) \\
18(x - 3^{-2}) & \text{if } x \in (3^{-2}, 1/6) \\
1 & \text{otherwise.}
\end{cases}
$$

The head operation works as a truncation to remove all the irrelevant bits in our output.

## B.3 Training via SGD

Let us now show how our construction performs memorization via SGD. In order to compute the gradient update, it will be helpful to define some new notation. For a matrix $X \in \mathbb{R}^{d_1 \times d_2}$, define a grid extractor function $\mathsf{grid} : X \times [d_1] \times [d_2] \to \mathbb{R}^{3 \times 3}$ such that $\mathsf{grid}(X, i, j)$ outputs the $3 \times 3$ sub-matrix of $X$ centered at $i, j$ with padding $p = 1/6$ at the edges. We will also define $\bar{f}_{\text{ConvLayer}}$ as the application of $\bar{f}_{\text{Conv2D}}$ to the entire input, that is, for input $X$, $\bar{f}_{\text{ConvLayer}}(X)_{i,j} = \bar{f}_{\text{Conv2D}}(\mathsf{grid}(X, i, j))$ for all $i, j$.

Figure 3: *(Left, Center)* $\bar{f}_{\text{Conv2D}}$ can identify the row to memorize in two ways: (left) At iteration 1, check if the current coordinate is 0 and the coordinate above is padding, and (center) at iteration $i \leq m+1$, check if current coordinate is 0 and coordinate above has bit 1 set, that is, has value $\pm 1/3$. Subsequently, $\bar{f}_{\text{Conv2D}}$ can update the coordinate value by adding $-3^{-2}$ to signal memorization phase to the neighbors. *(Right)* $\bar{f}_{\text{Conv2D}}$ can identify when memorization is done and set the head appropriately to start the TM roll-out: The left-bottom corner coordinate of the input (ignoring the padding) can identify its global position using the pattern around it, and check if it has memorized (by checking bit 1). Subsequently, $\bar{f}_{\text{Conv2D}}$ can update the coordinate value by adding $3^{-2} + 3^{-3}$ to signal computation phase to the neighbors and assign head location.

**Lemma 3** (Forward pass and backward gradient calculation). *For inputs $X \in \mathbb{R}^{(m+1)\times(d+1)}$, if there exists $1 \leq \tau \leq m+1$, such that $X_{i,j} \in \{\pm 3^{-1}\}$ for all $i < \tau, j \in [d+1]$ and $X_{i,j} = 0$ for all $i \geq \tau, j \in [d+1]$, then*

- *$\bar{f}^{(l)}_{\text{ConvLayer}}(Z)_{a,b} = -3^{-2}$ for all $a \in [m+1], b \in [d+1]$*

- *For all $i \in [m+1], j \in [d+1]$, $\nabla_{X_{i,j}} \bar{f}^{(l)}_{\text{ConvLayer}}(Z)_{a,b} = \begin{cases} 1 & \text{if } a = i = \tau, b = j \\ 0 & \text{otherwise} \end{cases}$*

*where $Z = [X \; \mathbb{1}_{(m+1)\times l}]$.*

*Proof.* Suppose the condition is satisfied for $\tau$, then we will prove by induction, the following claim: for $t > 0$, for all $b \in [d+1]$

$$\bar{f}^{(t)}_{\text{ConvLayer}}(Z)_{a,b} = \begin{cases} 3^{-2} & \text{if } |a - \tau| < t \\ Z_{a,b} & \text{otherwise.} \end{cases}$$

Note that this holds for $t = 1$:

$$\begin{aligned}
\bar{f}_{\text{ConvLayer}}(Z)_{\tau,b} &= \bar{f}_{\text{Conv2D}}(\text{grid}(Z, \tau, b)) \\
&= \bar{f}_{\text{msg}}(\text{grid}(Z, \tau, b)) & \textit{(since } X_{\tau,b} = 0\textit{)} \\
&= -3^{-2} & \textit{(first if condition of } \bar{f}_{\text{msg}}\textit{)}
\end{aligned}$$

For all $a \neq \tau$,

$$\begin{aligned}
\bar{f}_{\text{ConvLayer}}(Z)_{a,b} &= \bar{f}_{\text{Conv2D}}(\text{grid}(Z, a, b)) \\
&= \bar{f}_{\text{msg}}(\text{grid}(Z, a, b)) & \textit{(since } \text{val}(X_{a,b}, 2) = 0\textit{)} \\
&= Z_{a,b} & \textit{(no condition (1-4) is true of } \bar{f}_{\text{msg}}\textit{)}.
\end{aligned}$$

Let us assume it holds for $t$, and prove for $t + 1$. For $a$ such that $|a - \tau| < t$, since memory phase is set, $\bar{f}_{\text{Conv2D}}$ behaves like an identity match. For $a$ such that $|a - \tau| = t$, we have

$$\begin{aligned}
\bar{f}^{(t+1)}_{\text{ConvLayer}}(Z)_{a,b} &= \bar{f}_{\text{Conv2D}}(\text{grid}(\bar{f}^{(t)}_{\text{ConvLayer}}(Z), a, b)) \\
&= \bar{f}_{\text{msg}}(\text{grid}(\bar{f}^{(t)}_{\text{ConvLayer}}(Z), a, b)) & \textit{(since } \text{val}(X_{a,b}, 2) = 0\textit{)} \\
&= 3^{-2} & \textit{(second if condition of } \bar{f}_{\text{msg}}\textit{)}.
\end{aligned}$$

Similar to the base case argument, for all $a$ outside this band, none of the if conditions are satisfied and it acts as a pass through. Since $l > m + 1$, we get the first part of the above lemma.

For the second part, we will again prove by induction on depth of RCNN. For $l' = 1$, we have

$$\nabla_{X_{i,j}} \bar{f}_{\text{ConvLayer}}(Z)_{a,b}$$
$$= \nabla_{X_{i,j}} \bar{f}_{\text{Conv2D}}(\text{grid}(Z, a, b))$$
$$= \sum_{a',b' \in \{-1,0,1\}} \nabla_{a',b'} \bar{f}_{\text{Conv2D}}(\text{grid}(Z, a, b)) \cdot \nabla_{X_{i,j}} Z_{a+a',b+b'} \qquad \text{(by chain rule)}$$
$$= \nabla_{0,0} \bar{f}_{\text{Conv2D}}(\text{grid}(Z, a, b)) \cdot \nabla_{X_{i,j}} Z_{a,b} \qquad \text{(by gradient construction of } \hat{f}_{\text{Conv2D}})$$
$$= \nabla_{0,0} \bar{f}_{\text{Conv2D}}(\text{grid}(Z, a, b)) \cdot \mathbb{1}[a = i \wedge b = j].$$

Observe that, by if condition 2 of gradient of $\bar{f}_{\text{Conv2D}}$, for $a = \tau$, the gradient is 1. For all other coordinates, none of the if conditions are satisfied and hence the gradients are 0.

Let us assume for $l' < l$ and prove for $l' + 1$.

$$\nabla_{X_{i,j}} \bar{f}_{\text{Conv2D}}^{(l'+1)}(Z)_{a,b} = \nabla_{0,0} \bar{f}_{\text{Conv2D}}(\text{grid}(f_{\text{Conv2D}}^{(l')}(Z), a, b)) \cdot \nabla_{X_{i,j}} \bar{f}_{\text{Conv2D}}^{(l')}(Z)_{a,b}$$
$$= \nabla_{0,0} \bar{f}_{\text{Conv2D}}(\text{grid}(f_{\text{Conv2D}}^{(l')}(Z), a, b)) \cdot \mathbb{1}[a = i = \tau \wedge b = j].$$

From the induction before, we have that $\bar{f}_{\text{ConvLayer}}^{(l')}(Z)_{\tau,b} = 3^{-2}$ for all $l' > 0$, therefore by the gradient condition, we have $\nabla_{0,0} \bar{f}_{\text{Conv2D}}(\text{grid}(f_{\text{Conv2D}}^{(l')}(Z), \tau, b)) = 1$ giving us the desired result. $\square$

Let us now compute the back-propagated gradient for each input coordinate through the RCNN layers. Now we are ready to prove part 1 of Lemma 2 with $\bar{f}_{\text{Conv2D}}, \bar{f}_{\text{head}}$. We restate it as,

**Lemma 4.** *For $\tau < m + 2$ of Algorithm 1 with RCNN layers computed by $\bar{f}_{\text{Conv2D}}$, head computed using $\bar{f}_{\text{head}}$, and $\eta = 1/54$, we have:*

$$W_{ab}^{(\tau+1)} = \begin{cases} 3^{-1} y^{(a)} x_b^{(a)} & \text{if } a < \tau, b \leq d \\ 3^{-1} y^{(a)} & \text{if } a < \tau, b = d + 1 \\ 0 & \text{otherwise.} \end{cases}$$

*Proof.* We will prove by induction. Observe that, this trivially holds for $\tau = 0$ since $W^{(1)} = 0_{(m+1) \times (d+1)}$. Now let us assume it hold for $m + 1 > t > 0$, we will show that it holds for $t + 1$.

Since $x^{(t+1)} \in \{\pm 1\}^d$, we have $X^{(t+1)} = W^{(t+1)} \text{diag}\left(\left[x^{(t+1)\top} \; 1\right]^\top\right)$ is such that its first $t$ rows have entries in $\{\pm 3^{-1}\}$ and rest of the rows have entries 0. This satisfies the condition of Lemma 3 with $\tau = t + 1$, giving us the following:

- $\bar{f}_{\text{ConvLayer}}^{(l)}(Z^{(t+1)})_{a,b} = -3^{-2}$ for all $a \in [m+1], b \in [d+1]$

- For all $i \in [m+1], j \in [d+1], \nabla_{X_{i,j}} \bar{f}_{\text{ConvLayer}}^{(l)}(Z^{(t+1)})_{a,b} = \begin{cases} 1 & \text{if } a = i = t+1, b = j \\ 0 & \text{otherwise} \end{cases}$

where $Z^{(t+1)} = [X^{(t+1)} \; \mathbb{1}_{(m+1) \times l}]$.

Now let us compute the update step. Observe that $\bar{f}_{\text{head}}(-3^{-2}) = 0$, therefore $f(x^{(t+1)}; W^{(t+1)}, \bar{f}_{\text{Conv2D}}, \bar{f}_{\text{head}}) = 0$. Now, using chain rule and the above observations, we can compute the full desired gradient,

$$\nabla_{W_{a,b}^{(t+1)}} \ell\left(f(x^{(t+1)}; W^{(t+1)}, \bar{f}_{\text{Conv2D}}, \bar{f}_{\text{head}}), y^{(t+1)}\right)$$
$$= -(y^{(t+1)} - f(x^{(t+1)}; W^{(t+1)}, \bar{f}_{\text{Conv2D}}, \bar{f}_{\text{head}})) \nabla_{W_{a,b}^{(t+1)}} f(x^{(t+1)}; W^{(t+1)}, \bar{f}_{\text{Conv2D}}, \bar{f}_{\text{head}})$$
$$= -18 y^{(t+1)} \sum_{i,j} \nabla_{Z_{a,b}^{(t+1)}} \bar{f}_{\text{rc}}^{(l)}(Z^{(t+1)}) \cdot \nabla_{W_{a,b}^{(t+1)}} \bar{f}_{\text{mem}}(x^{(t+1)}; W^{(t+1)})_{i,j}$$
$$= -18 y^{(t+1)} \mathbb{1}[a = i] z_b^{(t+1)}.$$

where $z^{(t+1)} = [x^{(t+1)^\top} \; 1]^\top$. Note that the above follows from observing that the gradient of $\bar{f}_{\text{head}}$ at $-3^{-2}$ is 18 and the properties of the gradient of $\bar{f}_{\text{Conv2D}}$ from above.

With $\eta = 1/54$, we get that, $W^{(t+2)}$ satisfies the induction argument. This completes the proof. $\qquad\square$

Now once the memorization phase is over, the input to $\bar{f}_{\text{Conv2D}}$ will have on the tape, the training samples (in the XOR form with the current input) along with a clean input at the end (since $W_{m+1}^{(m+2)} = \mathbb{1}_{d+1}$ due to our dummy sample). At this stage, our $\bar{f}_{\text{Conv2D}}$ will implement $\bar{f}_{\text{msg}}$ to set the head (at $(m+1, d+1)$ position) using the third if condition. Once this is set, it will broadcast (using if condition 4) while the computation starts on those positions with computation flag set. Since our first step does not involve any movement of the head, the compute phase message passing protocol is always ahead of the head movement. Once the compute phase is set, $\bar{f}_{\text{Conv2D}}$ starts implementing $\bar{f}_{\text{TM}}$ which computes the roll-out of the TM. Finally, once the RCNN layers have been applied $\geq l + m + 1 + d$ times, the TM will terminate. The final layer output prior to $\bar{f}_{\text{head}}$ will be such that the top-left corner will have output set to either $\pm 3^{-1} + 3^{-2}$ (bit 1 will be set to $\pm 1$, bit 2 will be set to $3^{-2}$, and the rest of the bits being set to the final state), and the rest will be set at $3^{-2}$. $\bar{f}_{\text{head}}$ will ensure that this is truncated to $\pm 1$ accordingly. The rest of the coordinates in the final sum will be 0 since they will have value exactly $3^{-2}$ and will be truncated to 0 by $\bar{f}_{\text{head}}$. This proves the second part of Lemma 2.

Last we need to show that $\bar{f}_{\text{Conv2D}}$ and $\bar{f}_{\text{head}}$ can be constructed using NNs. It is not hard to see that $\bar{f}_{\text{head}}$ can be constructed using a one layer ReLU network with at most 6 ReLUs. To construct $\bar{f}_{\text{Conv2D}}$ requires not only function value matching on $\mathcal{V}_s$, but also gradient matching. The subsequent section proves a general representation representation result that allows us to do the same.

## B.4 Constructing $\bar{f}_{\text{Conv2D}}$ as a NN

Here show how the $\bar{f}_{\text{Conv2D}}$ can be implemented by 5-layer neural network in a way that keeps the parameters and their choices bounded by $\text{poly}(s)$. We show a general result that takes any function and gradient specification on discrete domains and converts it into a 5-layer network. Corollary 3 can be directly applied to our construction of $\bar{f}_{\text{Conv2D}}$ with $\mathcal{X} = \mathcal{Y}$ being all rational numbers with precision $O(\log(s))$ in base 3. This lemma gives us a construction of a 5-layer net where each parameter lies in a set of size $O(s)$. This set can be computed based on knowing the input domain, which is known a priori.

### B.4.1 Representing discrete functions with gradient pass-throughs

To implement the discrete functions used in the main construction, we make use of Lemma 5, which we state and prove in this section. It constructs a 5-layer fully-connected neural network whose values match those of an arbitrary multivariate real function on a finite domain, while allowing gradients to pass through for an arbitrary choice of inputs. The basic idea for the construction (build a basis of indicator functions, and enumerate over all possible input-output pairs) is not new, but we could not find an explicit theorem satisfying our additional requirements.[9] We hope this *function approximation lemma with custom gradients* will be useful beyond the scope of this paper. Specifically, beyond typical universal function approximation results, we need:

- Simultaneous control over the function values (all $d_{\text{out}}$ coordinates at all $|\mathcal{X}|$) and Jacobians (all $d_{\text{in}} \times d_{\text{out}}$ partial derivatives at all $|\mathcal{X}|$ points), so that the gradient signal from SGD can propagate through the recurrent computations to the memory layer in a controlled way.

- A bound on the number of distinct values the weights can take, so that we can analyze the probability that an i.i.d. random initialization scheme finds the desired weights.

To simplify notation, we will use a single matrix parameter $W^{d_{\text{out}} \times (d_{\text{in}}+1)}$ to parameterize an affine map $x \to W[x^\top \; 1]^\top$. Furthermore, we will overload notation and use the notation $W : \mathbb{R}^{d_{\text{in}}} \to \mathbb{R}^{d_{\text{out}}}$ to represent the same affine map.

---

[9] A proof sketch for the indicator construction can be found in (Nielsen, 2015). A quantitative version, without the additional considerations in this work, appears as Lemma B.8 in (Edelman et al., 2021), which requires a function representation lemma with robustness and weight norm bounds, for the purpose of obtaining non-vacuous generalization guarantees arising from sparsified inputs.

**Lemma 5.** *Let $\mathcal{X} \subset \mathbb{R}^{d_{\text{in}}}, \mathcal{Y} \subset \mathbb{R}^{d_{\text{out}}}, \mathcal{G} \subset \mathbb{R}$ be such that $|\mathcal{X}|, |\mathcal{Y}|, |\mathcal{G}| < \infty$. Let $f : \mathcal{X} \to \mathcal{Y}$, and let $g : \mathcal{X} \times [d_{\text{out}}] \times [d_{\text{in}}] \to \mathcal{G}$. Then, there exists a 3-layer ReLU network (letting $\sigma$ denote the entrywise ReLU function), with fully-connected layers specifying affine functions $W_1 : \mathbb{R}^{d_{\text{in}}} \to \mathbb{R}^{d_1}, W_2 : \mathbb{R}^{d_1} \to \mathbb{R}^{d_2}, W_3 : \mathbb{R}^{d_2} \to \mathbb{R}^{d_{\text{out}}}$, such that:*

$$f_{\text{net}}(x) := (W_3 \circ \sigma \circ W_2 \circ \sigma \circ W_1)(x) = f(x), \quad \forall x \in \mathcal{X}; \tag{1}$$

$$\frac{\partial}{\partial x_j}(W_3 \circ \sigma \circ W_2 \circ \sigma \circ W_1)(x)_i = g(x, i, j), \quad \forall x \in \mathcal{X}, i \in [d_{\text{out}}], j \in [d_{\text{in}}]. \tag{2}$$

*The intermediate dimensions satisfy*

$$d_1 = 10 d_{\text{in}} |\mathcal{X}|, \quad d_2 = 6 d_{\text{in}} d_{\text{out}} |\mathcal{X}|. \tag{3}$$

*Proof.* The construction consists of 3 steps:

(i) Build a univariate indicator (a bump function $\psi_{x_0} : \mathbb{R} \to \mathbb{R}$) for each input coordinate's domain $\mathcal{X}_i$, whose gradient is always 1 in the "bump" region, using a linear combination of 5 ReLU activations.

(ii) For any $x \in \mathcal{X}$ and $y, \gamma \in \mathbb{R}$, build multivariate indicators $\psi_{x_0, y}^{(\gamma)} : \mathbb{R}^{d_{\text{in}}} \to \mathbb{R}$, using a linear combination of 3 univariate indicators. We ensure that $\psi_{x_0, y}^{(\gamma)}$ has gradient $\gamma e_i$.

(iii) Assemble the function $f$ piece-by-piece: sum over indicators for each $(x, f(x), i \in [d_{\text{out}}])$, using the appropriate indicators to match every desired partial derivative $g(x, i, j)$.

We will implement all of the indicators from part (i) using linear combinations of ReLU activations $W_1 \circ \sigma \circ W_1'$, then the sum the indicators from part (ii) using $W_2' \circ \sigma \circ W_3$. The final network will simply compress the two intermediate affine functions into one: $W_2 = W_1' \circ W_2'$.

**Part (i).** Let $\Delta \leq \frac{1}{10} \min\left(1, \min_{x, x' \in \mathcal{X}, i \in [d_{\text{in}}]} |x_i - x_i'|\right)$. For a given $x_0 \in \mathbb{R}$ and desired gradient $\gamma \in \{0, 1\}$, let $\psi_{x_0}^{(\gamma)}$ denote the unique continuous piecewise linear function $\psi : \mathbb{R} \to \mathbb{R}$ such that:

- $\psi(x) = 0$ for $x \in (-\infty, x_0 - 2\Delta]$ and $x \in [x_0 + 2\Delta, \infty)$.
- $\psi(x_0) = 1$, and $\psi'(x) = \gamma$ for $x \in (x_0 - \Delta, x_0 + \Delta)$.
- $\psi'(x)$ is constant on $[x_0 - 2\Delta, x_0 - \Delta]$ and $[x_0 + \Delta, x_0 + 2\Delta]$.

By our choice of $\Delta$, for each $x_0, x \in \cup_{i \in [d_{\text{in}}]} \mathcal{X}_i$, $\psi_{x_0}^{(\gamma)}(x) = \mathbb{1}[x = x_0]$. In other words, $\psi$ is an indicator for a unique real number that appears in any coordinate of $\mathcal{X}$. Furthermore, $\psi_{x_0}^{(\gamma)\prime}(x) = \gamma \times \mathbb{1}[x = x_0]$. When $\gamma = 1$, $\psi$ lets the gradient pass through in the active region of the indicator; when $\gamma = 0$, $\psi$ blocks the gradient at every input in the domain.

Furthermore, $\psi$ consists of 5 linear regions, so it can be written as an affine function (i.e. linear combination, plus constant bias term) of 5 ReLU activations $W \circ \sigma \circ W'$, where $W : \mathbb{R} \to \mathbb{R}^5, W' : \mathbb{R}^5 \to \mathbb{R}$.[10] We construct $W_1, W_1'$ by concatenating the indicators $\psi_{x_0}^{(\gamma)}(e_i^\top x)$ for each $i \in [d_{\text{in}}], \gamma \in \{0, 1\}, x_0 \in \mathcal{X}_i$, so that $W_1' : \mathbb{R}^{d_1} \to \mathbb{R}^{d_1'}$, where $d_1' = 2 d_{\text{in}} |\mathcal{X}|$, the number of indicators we have constructed.

**Part (ii).** We will build a multivariate indicator $\Psi$ by summing per-coordinate univariate indicators, and checking that they sum to $d_{\text{in}}$. For a desired output $y$ and partial derivative $\gamma \in \mathcal{G}$, let $\tau_y^{(\gamma)}$ denote the unique piecewise linear function $\tau : \mathbb{R} \to \mathbb{R}$ such that:

- $\tau(x) = 0$ for $x \in (-\infty, d_{\text{in}} - 2\Delta]$.

---

[10] A sketch of this construction: for each discontinuity $\zeta$ of the desired piecewise linear function $\psi$, place a ReLU activation $\sigma(x - \zeta)$; also, for a value $\zeta_0$ less than all $\zeta$, place one more ReLU activation $\sigma(x - \zeta_0)$. Solve a linear system in the coefficients to get each linear region to agree with $\psi$.

- $\tau(d_{\text{in}}) = y$, and $\tau'(x) = \gamma$ for $x \in [d_{\text{in}} - \Delta, \infty)$.

- $\tau'(x)$ is constant on $(d_{\text{in}} - 2\Delta, d_{\text{in}} - \Delta)$.

By our choice of $\Delta$, the output of this function is $f(x_0)$ only when each summand is 1, which only happens when $x = x_0$. Otherwise, the input to $\tau$ is in the flat region, where $\tau(\cdot) = \tau'(\cdot) = 0$. In summary, $\tau$ is an indicator like $\psi$. It is slightly simpler to construct, since it only needs to implement a threshold function, and only needs to recognize that its input is $d_{\text{in}}$ (rather than an arbitrary $x_0$). Since each $\tau$ has 3 linear regions, it can be built with an affine function of 3 ReLU activations.

Our network will construct $d_2' = 2d_{\text{in}}d_{\text{out}}|\mathcal{X}|$ of these indicators:

$$\tau_y^{g(x_0,i,j)} \quad \forall x_0 \in \mathcal{X}, i \in [d_{\text{in}}], j \in [d_{\text{out}}], y \in \{0, f(x_0)_j\}.$$

We set $W_2'$ to be the concatenation of all of these indicators, so that $W_2' : \mathbb{R}^{d_2} \to \mathbb{R}^{d_2'}$.

**Part (iii).** Then, for each $x_0 \in \mathcal{X}$ and $j \in [d_{\text{out}}]$, with corresponding desired function values $f(x_0)_i$ and partial derivatives $g(x_0, i, j)$, we construct the indicator for a single output coordinate:

$$\Psi_{f,g,x_0,j}(x) := \sum_{i'=1}^{d_{\text{in}}} \tau_{f(x_0)_j \cdot \mathbb{1}[i'=1]}^{(g(x_0,i',j))} \underbrace{\left( \sum_{i=1}^{d_{\text{in}}} \underbrace{\psi_{(x_0)_i}^{(\mathbb{1}[i=i'])}(e_i^\top x)}_{\text{computed by } W_1 \circ \sigma \circ W_1'} \right)}_{\text{computed by } W_1 \circ \sigma \circ W_2 \circ \sigma \circ W_2'} \tag{4}$$

The intuition behind Equation 4 is the following: if we only cared about matching the function value $f$ at all points in $\mathcal{X}$, it would suffice to use one indicator per $x_0 \in \mathcal{X}, j \in [d_{\text{out}}]$. However, we need to get every coordinate of the gradient correct. To implement this, we create $d_{\text{in}}$ redundant indicators for each $x_0$, and sum over all of them, ensuring that the gradient is counted once per input coordinate, and the function value is counted once in total. Finally, $W_3$ is constructed by summing over all of the indicators from the previous part:

$$f_{\text{net}}(x)_j = \sum_{x \in \mathcal{X}} \Psi_{f,g,x_0,j}(x), \quad \forall j \in [d_{\text{out}}].$$

Lines 1 and 2 in the statement follow from the inline discussions above of properties of the indicator modules. □

Next, we will bound the size of the support of i.i.d. random initialization weights needed to construct the network in Lemma 5. To do this with fewer distinct values, we will make the following changes to the architecture:

- Split the layer $W_2$ into the composition of two affine layers $W_1' \circ W_1^+$, with intermediate dimension $d_1'$, according to the above analysis.

- Similarly, split the layer $W_3$ into $W_2' \circ W_2^+$, with intermediate dimension $d_2'$.

Overall, this expands the 3-layer 2-ReLU architecture to a 5-layer 2-ReLU equivalent, changing the parameterization but not the class of representable functions. With this modified construction, we show that the unique nonzero matrix weights for the linear layers $W \in \{W_1, W_1', W_1^+, W_2', W_2^+\}$ lie in a bounded-size domain $\mathcal{U}(W)$, which only depends on $\mathcal{X}, \mathcal{Y}, \mathcal{G}$, not $f, g$. These $\mathcal{U}$ allow us to define the support of the random initialization distribution, and determine the probability of success. We state and prove the bounds on $|\mathcal{U}(\cdot)|$ below.

**Lemma 6.** *Let $W_1(f, g), W_1'(f, g), \ldots$ be the matrices arising from the modified construction from Lemma 5. Then, there exist finite sets $\mathcal{U}(W) \subset \mathbb{R}$ depending only on $\mathcal{X}, \mathcal{Y}, \mathcal{G}$, such that $\mathcal{U}(W) \cup \{0\}$ contains all elements $W(f, g)$, and:*

- $|\mathcal{U}(W_1)| \le 4|\mathcal{X}_i| + 2.$

- $|\mathcal{U}(W_1')| \le 12|\mathcal{X}_i|.$

- $|\mathcal{U}(W_1^+)| \le 4.$

- $|\mathcal{U}(W_2')| \leq 4|\mathcal{G}|\,|\mathcal{Y}_i|$,
- $|\mathcal{U}(W_2^+)| = 1$.

*where $\mathcal{X}_i := \{x_i : x \in \mathcal{X}, i \in d_{\text{in}}\}$ and $\mathcal{Y}_i := \{x_i : x \in \mathcal{Y}, i \in d_{\text{out}}\}$ denote the sets of possible input and output values.*

*Proof.* First, notice that with a known, finite $\mathcal{X}$, there is a deterministic way to choose a sufficiently small $\Delta$, and a sufficiently large $\zeta_0$ (the lowest bias in the ReLU-to-piecewise linear constructions). We analyze the construction of each layer, and enumerate the possible nonzero weights and biases:

- It is clear from the construction that the linear coefficients in $W_1$ are in $\{0, 1\}$. Furthermore, for each $x_0$ occurring in any coordinate of an element in the domain $\mathcal{X}$, there are 5 bias terms: $-\zeta_0, -x_0 \pm \Delta, -x_0 \pm 2\Delta$.

- $W_1'$ maps groups of 5 ReLU activations to the corresponding indicators $\psi$. The coefficients are each functions of a single $x_0$ and $\Delta$; there are 6 coefficients (including one bias) per indicator, and 2 indicators ($\gamma \in \{0, 1\}$) per $x_0$.

- $W_1^+$ combines the indicators $\psi$ to form the inputs to the ReLUs, which $W_2'$ will use to form the indicators $\tau$. These weights are again in $\{0, 1\}$, and biases are from the ReLU discontinuity locations: $-\zeta_0, -(d_{\text{in}} - 2\Delta), -(d_{\text{in}} - \Delta)$.

- $W_2'$ forms the $d_2'$ indicators $\tau$, with 4 coefficients per indicator, depending on $\mathcal{G}, \mathcal{Y}, \Delta, d_{\text{in}}$.

- $W_2^+$ simply takes a summation over the indicators $\tau$, so its coefficients are in $\{0, 1\}$.

$\square$

We summarize the results in Lemma 6 with a looser corollary:

**Corollary 3.** *Let $\mathcal{X}, \mathcal{Y}, \mathcal{G}$ be known and finite, and let $\mathcal{X}_i = \mathcal{Y}_i$, $\mathcal{G} = \{0, 1\}$. Using the construction of $\mathcal{U}(\cdot)$ in Lemma 6, we can define a single set*

$$\mathcal{U} := \{0\} \bigcup_{W \in \{W_1, W_1', W_1^+, W_2', W_2^+\}} \mathcal{U}(W),$$

*which contains all possible weights and biases in all layers of the ReLU network. The cardinality of $\mathcal{U}$ satisfies*

$$|\mathcal{U}| \leq 24|\mathcal{X}_i| + 5.$$

Given this corollary, for our setting, $\mathcal{U}_s$ has size $24 \cdot 16 \cdot s + 5$, and is described above, can be constructed with knowledge of only $s$.

### B.5 Proof of Corollary 2

Let $M$ be the number of random restarts and $N$ be the size of the validations set, Using standard concentration bounds, we know that with probability $1 - \delta/2$, the expected error $\text{err}_{\mathcal{D}}$ of classifiers for all $f_1, \ldots, f_M$ over the distributions is within $\sqrt{\frac{\log(2M/\delta)}{2N}}$ of the empirical error $\hat{\text{err}}_{\mathcal{D}}$. This implies that the error of the classifier $\hat{f}$ selected by our validation set satisfies

$$\text{err}_{\mathcal{D}}(\hat{f}) \leq \hat{\text{err}}_{\mathcal{D}}(\hat{f}) + \sqrt{\frac{\log(2M/\delta)}{2N}}$$

$$\leq \min_{i \in [M]} \hat{\text{err}}_{\mathcal{D}}(f_i) + \sqrt{\frac{\log(2M/\delta)}{2N}}$$

$$\leq \min_{i \in [M]} \text{err}_{\mathcal{D}}(f_i) + 2\sqrt{\frac{\log(2M/\delta)}{2N}}$$

By Theorem 1, we know that with probability $1 - Ms^{c_2 s^2}$, at least one (say $f_1$) of the $f_1, \ldots, f_M$ will satisfy the $\mathrm{err}_{\mathcal{D}}(f_1) \leq \mathrm{err}_{\mathcal{D}}(A, \mathcal{S})$. This gives us, with probability $1 - \delta/2 - Ms^{c_2 s^2}$, we have,

$$\mathrm{err}_{\mathcal{D}}(\hat{f}) \leq \mathrm{err}_{\mathcal{D}}(A, \mathcal{S}) + 2\sqrt{\frac{\log(2M/\delta)}{2N}}.$$

Setting $M, N$ such that $2\sqrt{\frac{\log(2M/\delta)}{2N}} = \epsilon$ and $Ms^{-c_2 s^2} = \delta/2$, we get the desired result. $M = \delta s^{c_2 s^2}/2$ and $N = \frac{c_2 s^2 \log(s/2)}{8\epsilon^2}$.