# OpenReview forum: "Recurrent Convolutional Neural Networks Learn Succinct Learning Algorithms"
_NeurIPS.cc/2022/Conference — NeurIPS 2022 Accept_

### Official Review · Reviewer_UfCN · 2022-07-12

**Rating:** 5
**Confidence:** 3
**Soundness:** 3 good
**Presentation:** 3 good
**Contribution:** 2 fair

**Summary:**

This paper proposes a notion of turing-optimal learning, which means that a learning algorithm can learn as well as any Turing Machine with s states, m examples, and that terminates in t steps. A neural network with convolution and recurrent layers are designed. This network has only constant number of distinct parameters, but the depth and width grow polynomially in data dimension d, sample size m, and number of turing states s. Using repeated random restart and gradient descent, an algorithm that learns with this architecture is shown to be turing-optimal.

**Questions:**

1. Line 97, did you mean [Wei et al.] doesn't take computational efficiency into account?
2. Can you elaborate how random restarts are different from enumerative program search?
3. Typo in line 272?

**Limitations:**

Yes.

**Strengths And Weaknesses:**

Strengths:
1. The notion of Turing-Optimal Learning is original, intuitive, and interesting.
2. The theoretical part is rigorous and requires significant effort.
3. The exposition is clear, and the proof sketch is informative.

Weaknesses:
The construction is far from practical (and the authors admitted as much). SGD plays nominal part in the picture; the network basically memorizes all training examples, and then uses $s^s$ random guesses for the correct algorithmic parameters (via random restart). Overall constructions like this do not meaningfully inform ML practitioners, as it's unlikely to memorize all training examples and do random guessing in practice, and it's also undesirable that the network size grows with sample size, and undetermined when s isn't known for realistic datasets.

---

> ### Author Response · Authors · 2022-08-02
> **Response to Reviewer UfCN**
>
> First, note that the purpose of our construction is to show that the architecture is capable of learning algorithms in polynomial time– it is not intended to be practical as is. However, we do believe that recurrent/convolutional weight sharing and a memory component may be useful for designing architectures that emulate algorithms. Furthermore, recent work [1] on generalization to larger inputs/ harder problems use recurrent weight sharing to learn the underlying algorithm and implement extra roll-outs (reapplications of the recurrent layer) to simulate more computation. This is similar to our use of depth, where depth is indicative of the runtime of the learning algorithm. We refer the reviewer to Section 5 in the paper for a more detailed discussion. As for the specific concerns:
>
> *"Memorize all training examples and do random guessing in practice:"* We agree that this seems like an oversimplified view of how learning may be happening, however there is evidence that several architectures do memorize training data, for example, large language models can be attacked to produce exact training sequences [2]. Furthermore, random guessing (or exhaustive search) do show up in different parts of the training pipeline, for example, whole performing hyperparameter search, learning rate tuning, etc. More importantly, our construction only shows that this naive algorithm is implementable by our pipeline. In practice, we could hope for SGD training to potentially find better solutions.
>
> *"Network size grows with sample size:"* We do understand the concern about network size growing with input dimension and number of samples, however, current models do seem to be operating in this regime where the best performing architectures have a number of parameters much larger than input dimension and number of training samples.
>
> *"Undetermined when $s$ isn't known:*" In the case of unknown $s$, a simple approach is to start with some value of $s$ and then use a validation set to decide whether to increase the estimate of $s$.
>
> **Responses to Questions:**
>
> 1. (Wei et al.) consider the notion of statistically meaningful approximations of Turing Machines, in which they combine statistical efficiency (in terms of number of samples) and representation (approximability by the function class). They do not account for whether the underlying function class is computationally learnable, that is, if there is a polynomial time algorithm that can learn the underlying function class. In our setting, we care about computational efficiency of the architecture when trained via SGD.
>
> 2. They are not different from each other in terms of their purpose. However, unlike enumerative program search, which may not be easy to actually implement, our construction instead does this through random restarts. Furthermore, since our networks are trained via SGD, we can potentially get better solutions. Our theory only argues that there is a good solution given enough random restarts.
>
> 3. Thank you for pointing out the typo, we have corrected it.
>
> [1] Schwarzschild, A., Borgnia, E., Gupta, A., Huang, F., Vishkin, U., Goldblum, M., Goldstein, T.. (2021). Can You Learn an Algorithm? Generalizing from Easy to Hard Problems with Recurrent Networks. arXiv preprint arXiv:2106.04537.
>
> [2] Carlini, N., Tramer, F., Wallace, E., Jagielski, M., Herbert-Voss, A., Lee, K., Roberts, A., Brown, T., Song, D., Erlingsson, U., et al. (2020). Extracting training data from large language models. arXiv preprint arXiv:2012.07805.

---

> > ### Comment · Reviewer_UfCN · 2022-08-08
> > **Thank you for your response**
> >
> > I thank the authors for their response. I like how clearly this paper is written, which is uncommon for approximation theory papers, even though it is not surprising to me that memorizing the training set + randomly guessing the parameters $s^s$ times would work which is similar to exhaustive program search. I'm raising my score to borderline accept.

---

### Official Review · Reviewer_AW8a · 2022-07-12

**Rating:** 7
**Confidence:** 4
**Soundness:** 3 good
**Presentation:** 3 good
**Contribution:** 3 good

**Summary:**

This paper analyzes recurrent neural networks' ability to learn algorithms. The authors show that the proposed architecture is Turing complete. They also show that the recurrent convolutional models discussed can learn parity functions in polynomial time.

**Questions:**

1. In the empirical work in this space, there are many popular, well-known, algorithms considered. What baring does the theory presented in this paper have on recurrent neural networks' ability to learn in settings outside parity?
2. In Section 5, the authors state that the constants in the theorems are "too large to be meaningful in practice." I appreciate that this is mentioned and I wonder what, if anything, can be done or where we might look to shrink those constants. Is this a theoretical limitation or an observation about the analysis in this particular paper?

**Limitations:**

The limitations are adequately addressed.

**Strengths And Weaknesses:**

Strengths
- To the best of my knowledge this is an original treatment of recurrent convolutional networks. This touches on an increasingly important field of algorithm learning and reasoning in neural networks.
- The writing is clear.
- The results in this paper are likely interesting to the community. Specifically, much of the recent work on algorithm learning is empirical, and theoretical treatment accompanies this well.
- Additionally, the theoretical grounding for choosing recurrence/weight sharing of this type might motivate the applications-forward researchers as well.

Weaknesses
- As is mentioned by the authors, the bounds/constants are not practical.
- Given the body of work (cited by the authors) on algorithm learning, it would be nice if there was a little more context as to the scope of the results in this paper. (See the questions below.)

---

> ### Author Response · Authors · 2022-08-02
> **Response to Reviewer AW8a**
>
> We thank the reviewer for their encouraging evaluation. Our study is meant to be of a theoretical nature where our main contribution is to show the existence of a natural NN training pipeline that can be run in polynomial time to learn constant sized algorithms. In Section 5, we have included a more detailed discussion of the potential practical implications of our work.
>
> 1. Our results are not restricted to parity; we just refer to it for ease of explanation. The construction holds for any class of algorithms that can be represented as a constant sized Turing Machine.
> 2. The constants being large is an artifact of our construction. One could hope for a better construction that improves on the constants. For example, we could hope to get more from the SGD updates that assist in our algorithm search reducing the need to restart a large constant number of times.
>
> **Regarding the bearing of these theoretical results on the practice of learning RNNs**: we believe that exhaustive search over concise parameterizations may be a necessary technique for challenging algorithmic and reasoning tasks requiring generalization across input lengths. We hope that this work will contribute to the theoretical foundations of how to reconcile gradient-based training with exhaustive search, as this field of “learning algorithms with neural networks” matures. Note that exhaustive search (not over network weights) already appears during inference, in the form of beam search and chain-of-thought generation.

---

> > ### Comment · Reviewer_AW8a · 2022-08-07
> > **Follow up**
> >
> > Thank you for the answers to my questions. I have read the other reviewers' feedback as well. I like the work and I kindly disagree with the only other review with high confidence. The fairly young area of algorithm learning is likely to benefit from theoretical work even if it may seem impractical to some. I support accepting this paper. I have updated my score.

---

### Official Review · Reviewer_Jx5f · 2022-07-12

**Rating:** 5
**Confidence:** 1
**Soundness:** 3 good
**Presentation:** 3 good
**Contribution:** 2 fair

**Summary:**

This paper introduces the notion of Probably Algorithmically Optimal (PAO) learning as a weaker notion than PAC learning, and the concept of a (meta?) learning algorithm being Turing-optimal if it can discover learning algorithms producing classifiers that perform as well as any classifier that can be obtained by another learning algorithm. They illustrate that a particular architecture complies with such a notion.

**Questions:**

1) Section 1, line 36: Do we really know that "two-layer NNs can compute any Boolean function on d binary inputs? I understand that the following references [1-2] are not restricted to Boolean domains, but I would appreciate an explanation for this statement.

[1] https://arxiv.org/abs/1611.01491

[2] https://proceedings.neurips.cc/paper/2021/hash/1b9812b99fe2672af746cefda86be5f9-Abstract.html

2) Is it reasonable to expect |A| << |C| as mentioned in line 61? Wouldn't it be more likely that the opposite is true, given that each algorithm in A is distinguished from other algorithms by producing a mapping of datasets to classifiers in C?

3) Can you provide an example for the claim in lines 129-130 that "a classifier may be very slow to evaluate, even if the learning algorithm is fast"? Likewise, under what circumstance would the case in lines 133-134 that "time spent on classification is folded into training time" be useful in practice?

4) Isn't it atypical to assume a fully connected layer before a sequence of convolutional layers? Why is this construction needed, and what practical relevance would it have to design architectures in such a way?

5) Likewise, isn't it limiting to assume recurrent weight sharing?

6) My understanding from the first sections is that the strict parameter sharing in the convolutional layers was meant to make the learning algorithm independent of input size, but the initial dense layer breaks that. What am I missing here?

7) Is the output of Algorithm 1 a classifier, a learning algorithm aimed at learning a classifier for a specific dataset, or a generic learning algorithm?

**Limitations:**

The authors are upfront about the practical limitation of their algorithm for realistic cases due to the large constants even if the algorithms are polytime.

**Strengths And Weaknesses:**

The paper is reasonably well written and the authors clearly position their proposed notions with respect to other theoretical models on learning algorithms. Given that this is not my area of expertise, at some point the idea that their model is supposed to produce a learning algorithm rather than a classifier gets more difficult to follow. However, my main concern - which may or may not be valid - is whether the RCNN that they study has any practical insight, given that its architecture is not something I would have expected in practice. Perhaps related to that, it is not entirely clear to me if these new notions (PAO and Turing-optimal) are practical or particularly relevant for further studies.

Minor comments on writing:
- Line 41: "innovation" -> contribution?
- Line 52: "to use a use a": remove "a use"
- Line 180: "the a number of examples": remove "a"
- Line 251: "1can" -> "1 can" (add space)
- Line 254: Please repeat that m is the sample size and d is the input size
- Line 272: It is not immediate what the three hyperparameters at the end of the sentence mean; like above, it would be very helpful to repeat that information next to each hyperparameter

---

> ### Author Response · Authors · 2022-08-02
> **Response to Reviewer #Jx5f**
>
> We thank the reviewer for their thoughtful review and for pointing out typos. We have corrected the typos in the new version of the paper. Below we address the concerns raised by the reviewer:
>
> 1. One way to see how we could implement any boolean function using a one-layer ReLU network is the following:
> - Decompose the function $f$ into its fourier decomposition, that is, $f(x) = \sum_{S \subseteq [d]} \hat{f}_S \chi_S(x)$ where $\chi_S(x)$ is the parity computed on set $S$ .
> - Implement each $\chi_S(x)$ using a sum of ReLUs (see Appendix C in [1]).
> - Implement the entire function by appropriately multiplying each $\chi_S(x)$ with $\hat{f}_S$.
>
> 2. We do believe that it is reasonable to assume that $|A| \ll |C|$ (of all the assumptions made in this paper, this seems to be the most benign). The number of parameters of the classifier is $O(\log |C|)$, and modern classifiers have billions or even trillions of parameters. The file size (in bits when compressed) of the learning program is $O(\log |A|)$. There have been many dramatic examples of a few-line programs that learn excellent but large neural networks. The fact that $|A| \ll |C|$ is discussed in great detail in, e.g., https://arxiv.org/abs/2102.13189.
>
> 3. We have clarified in (a) Natural examples where inference is more computationally expensive than learning arise in nonparametric models such as nearest-neighbors or Gaussian processes. (b) What we are saying here is that if we are concerned with efficient training and inference, then it does not matter how we divide up the time. This point is not about practice--it is for simplifying the analysis (which is already quite complex) to show that various theoretical models are equivalent from a polytime perspective.
>
> 4. The reviewer is correct, it is atypical to have convolutions after a fully connected layer. However it is needed to implement memory in our construction. Unlike, in vision where the locality is expected in the input space, our convolution serves a different purpose, where it intends to implement the locality of the learning algorithm (or the Turing Machine). The fully connected layer embeds the input to a space where the locality is reasonable.
>
> 5. If you take the perspective of representation, then in that sense allowing different parameters in every layer may limit the total number of possible classifiers. However, what’s interesting in our setting is that, if we view the network as learning an algorithm, then recurrent weight sharing is helpful to reduce parameter count without loss of computational power.
>
> 6. Note that the algorithm that runs on $m$ samples of dimension $d$ each would need to process these samples in its memory. The dense layer is intended to exactly do that and is zero initialized and trained via SGD. The only parameters that random restarts need to identify are the convolutional parameters which are independent of the input dimension and sample size.
>
> 7. Our Algorithm 1 outputs a classifier which is the output of the learning algorithm $A$ on the training set $\mathcal{S}$.
>
> [1] Mukherjee, A., & Basu, A. (2017). Lower bounds over boolean inputs for deep neural networks with relu gates. arXiv preprint arXiv:1711.03073.

---

> > ### Comment · Reviewer_Jx5f · 2022-08-07
> > **Following up**
> >
> > I would like to thank the reviewers for their careful feedback. Although I liked the paper and gave it a score of 5 despite this paper being outside my comfort zone, I concur with the comments of other reviewers that the setting studied is not very practical, including from the only reviewer with a confidence of 3. For that reason, I will keep my score intact and defer to the expertise of other PC members on the area.

---

### Official Review · Reviewer_ZUm5 · 2022-07-13

**Rating:** 5
**Confidence:** 1
**Soundness:** 2 fair
**Presentation:** 2 fair
**Contribution:** 2 fair

**Summary:**

The paper describes a theoretical model based on recurrent and convolutional neural modules that achieves what the authors call "Turing-optimality".
More specifically, the authors argue that a randomly initialized recurrent and convolutional NN that they introduce, trained with SGD with random restarts is probably algorithmically optimal for the class of algorithms given by constant-sized time-bounded Turing machines.`

**Questions:**

See above

**Limitations:**

Discussed

**Strengths And Weaknesses:**

Unfortunately, I am very unfamiliar with the literature and problems of this area and can only evaluate this paper in a very shallow manner.
I found the paper quite unclear and rather hard to follow. It seems to jump to conclusions before building preliminary concepts and sufficient intuition.
Some passages are rather unclear and seem to be contradictory: eg :
*The contribution of this work is showing that a definition
of Turing-optimality is achievable by a simple NN architecture. In future work, it would be interesting
to better understand which architectures, initializations, and learning rates are Turing-optimal.*
I am left to wonder whether one contribution of this paper is indeed that mentioned in the first reported sentence or not.
There are missing links to equations (references to equations point at sections, probably) and a rather free usage of Claims and Observations which make it hard to understand what are the assumptions that the authors do at any point of the paper.

My overall impression is that this papers lack (at the very least) in clarity and coherency, yet because of my lack of expertise in this field, I will only rate it as a weak reject.

---

> ### Author Response · Authors · 2022-08-02
> **Response to Reviewer #ZUm5**
>
> We thank the reviewer for acknowledging that this is not their area of expertise, and for giving us useful feedback on improving our writing. We have corrected the missing equation links. Below we address the reviewer's concerns:
>
> >The contribution of this work is showing that a definition of Turing-optimality is achievable by a simple NN architecture. In future work, it would be interesting to better understand which architectures, initializations, and learning rates are Turing-optimal.
>
> We apologize for this typo on a crucial point, and understand the apparent contradiction. We meant to say: “...it would be interesting to better understand which **other combinations of** architectures, initializations, and learning rates are Turing-optimal.
>
> To clarify, our **main contribution** is indeed to design a theoretical neural network training pipeline that achieves Turing-optimality. Our pipeline uses a recurrent convolutional network, a dense memory layer, and SGD with random restarts for training. There are certain elements of our pipeline that do not completely match practice, for instance, our random initialization is from a carefully-chosen discrete distribution, and our learning rate is set to 0 for part of the network. We hope that future work can help understand what parts of the pipeline are essential and what parts can be brought closer to their current practical implementations.

---

> > ### Comment · Reviewer_ZUm5 · 2022-08-07
> > **Reply**
> >
> > I thank the authors for their reply. I symbolically raise my score to 5, but defer to the other reviewers and AC.

---

### Meta-Review · Area_Chair_8bx6 · 2022-08-24

**Recommendation:** Accept
**Confidence:** Certain

**Metareview:**

In discussion of this paper, we agree that this paper presents a very novel treatment of learning algorithms and presents a new topic with an original formalization of the problem, which will be of substantial interest to the NeurIPS community.
The reviewers agreed that a formal understanding of RNN is of interest, it is not clear how close is the given framework to reality, and how much insight can be gained: The bounds are huge for practical datasets, and the use if SGD is minimal.
Overall, the significant novelty for the formulation of the question itself warrants acceptance.

**Award:**

Yes

---

### Decision · Program_Chairs · 2022-09-14

Accept